# Disentangling the Ecological Processes Shaping the Latitudinal Pattern of Phytoplankton Communities in the Pacific Ocean

Zhimeng Xu,[a,b,c,d] Shunyan Cheung,[d] Hisashi Endo,[e] Xiaomin Xia,[f,g] Wenxue Wu,[h] Bingzhang Chen,[i,j] Ngai Hei Ernest Ho,[d] Koji Suzuki,[k] Meng Li,[a,b] Hongbin Liu[c,d,g,j]

[a]SZU-HKUST Joint PhD Program in Marine Environmental Science, Shenzhen University, Shenzhen, China

[b]Shenzhen Key Laboratory of Marine Microbiome Engineering, Institute for Advanced Study, Shenzhen University, Shenzhen, China

[c]Southern Marine Science and Engineering Guangdong Laboratory (Guangzhou), Guangzhou, China

[d]Department of Ocean Science, The Hong Kong University of Science and Technology, Hong Kong, China

[e]Bioinformatics Center, Institute for Chemical Research, Kyoto University, Kyoto, Japan

[f]Key Laboratory of Tropical Marine Bio-resources and Ecology, South China Sea Institute of Oceanology, Chinese Academy of Sciences, Guangzhou, China

[g]CAS-HKUST Sanya Joint Laboratory of Marine Science Research, Sanya, China

[h]School of Marine Sciences, Sun Yat-sen University, Zhuhai, China

[i]Department of Mathematics and Statistics, University of Strathclyde, Glasgow, United Kingdom

[j]Department of Ocean Science and Hong Kong Branch of the Southern Marine Science and Engineering Guangdong Laboratory (Guangzhou), The Hong Kong University of Science and Technology, Hong Kong, China

[k]Graduate School of Environmental Science, and Faculty of Environmental Earth Science, Hokkaido University, Sapporo, Hokkaido, Japan

**ABSTRACT** Phytoplankton diversity and community compositions vary across spaces and are fundamentally affected by several deterministic (e.g., environmental selection) and stochastic (e.g., ecological drift) processes. How this suite of different processes regulates the biogeography of phytoplankton remains to be comprehensively explored. Using high-throughput sequencing data and null model analysis, we revealed the ecological processes shaping the latitudinal community structure of three major phytoplankton groups (i.e., diatoms, *Synechococcus*, and haptophytes) across the Pacific Ocean (70°N, 170°W to 35°S, 170°W). At the basin scale, heterogeneous selection (selection under heterogeneous environmental conditions) dominated the assembly processes of all phytoplankton groups; however, its relative importance varied greatly at the climatic zonal scale, explaining the distinct latitudinal $\alpha$- and $\beta$-diversity among phytoplankton groups. Assembly processes in *Synechococcus* and haptophyte communities were mainly controlled by physical and nutrient factors, respectively. High temperature drove *Synechococcus* communities to be more deterministic with higher diversity, while haptophyte communities were less environmentally selected at low latitudes due to their wide niche breadth and mixotrophic lifestyle. Diatom communities were overwhelmingly dominated by the selection process but with low correlation of measured environmental factors to their community compositions. This could be attributed to the high growth rate of diatoms, as indicated by their lower site occupation frequency than predicted in the neutral community model. Our study showed that heterogeneous selection is the main force that shaped the biogeography of three key phytoplankton groups in the Pacific Ocean, with a latitudinal variation of relative importance due to the distinct traits among phytoplankton.

**IMPORTANCE** Phytoplankton are diverse and abundant as primary producers in the ocean, with diversity and community compositions varying spatially. How fundamental processes (e.g., selection, dispersal, and drift) regulate their global biogeography remains to be comprehensively explored. In this study, we disentangled the ecological processes of three key phytoplankton groups (i.e., diatoms, *Synechococcus*, and haptophytes) along the same latitudinal gradients in the Pacific Ocean. Heterogeneous selection, by promoting species richness and reducing similarity between communities, was the dominant process shaping the communities of each phytoplankton group at the basin scale. However, its

Address correspondence to Hongbin Liu, liuhb@ust.hk.

The authors declare no conflict of interest.

relative importance varied greatly among different phytoplankton groups in different climate zones, explaining the uneven latitudinal $\alpha$- and $\beta$-diversity. We also highlight the importance of identifying key factors mediating the relative importance of assembly processes in phytoplankton communities, which will enhance our understanding of their biogeography in the ocean and future patterns under climate changes.

**KEYWORDS** Pacific Ocean, ecological process, latitudinal biogeography, phytoplankton community, spatial scale

Phytoplankton, including both prokaryotes (e.g., cyanobacteria) and eukaryotes (e.g., diatoms, haptophytes, and dinoflagellates), are diverse primary producers in the ocean. The dynamics of their community structure may have great consequences for global biogeochemical cycles (1). Due to their tiny size, complex cellular characteristics, and high diversity, community compositions of phytoplankton are hard to fully characterize by traditional methods (e.g., microscopy). Advances in sequencing technology (e.g., high-throughput sequencing) enable scientists to recover the microbial community structures from environmental DNA samples with a high coverage of diversity in the global oceans (2–5). Global diversity and distribution of phytoplankton have been extensively reported, and some ecological processes (e.g., selection by local environment and dispersal by ocean current) act as fundamental mechanisms regulating phytoplankton community structure (6–8). For instance, temperature mainly drives the present and future latitudinal patterns of marine phytoplankton both directly (e.g., enhancing speciation and metabolic rates) and indirectly (e.g., changing stratification, circulation, and trophic interactions), while dispersal ability can also play important roles in determining the biogeography of some phytoplankton groups (e.g., diatoms) (8–13).

According to the framework proposed by Vellend, ecological processes that shape the community assemblages (i.e., metacommunity) can be classified into four fundamental types: selection, dispersal, ecological drift, and speciation (14, 15). Selection is a niche-based process due to fitness differences (e.g., survival and growth) among organisms including both abiotic (e.g., chemical and physical factors) and biotic interactions (e.g., competition and predation). Selection can act in opposite directions by reducing (homogeneous selection) or increasing (heterogeneous selection) the diversity of communities due to environmental conditions. Dispersal is the movement of organisms across space, with a low level of dispersal leading to dispersal limitation and a high level of dispersal leading to homogenizing dispersal. Ecological drift (referred to as here as drift) is the random changes in species' relative abundance caused by stochastic death, birth, and immigration. Last, speciation is the creation of new species by genetic mutation, and it is not considered here due to its small impact on the metacommunity connected via dispersal (16–18).

Vellend's conceptual framework has gained popularity in the study of microbial community, as it defines the process at a finer level and systematically takes both deterministic processes (also known as niche processes, e.g., selection) and stochastic processes (also known as neutral processes, e.g., ecological drift) into consideration, both of which are crucial for modeling microbial ecology (19). Following this, the fundamental mechanisms regulating the community assembly can be reflected by estimating the relative contribution of each ecological process by different approaches (17, 20). So far, studies quantifying the ecological processes in marine microbial communities have focused mostly on bacteria and protists (21–23), while the mechanisms shaping the phytoplankton communities (especially some key groups) remain unclear from the bulk communities. In particular, different phytoplankton groups (or taxa) have distinct latitudinal diversity, abundance, and community structure (24–26); however, whether these biogeographic patterns are structured by the action of the same or different ecological processes is much less studied.

In this study, we characterized the phytoplankton community structures in the pelagic Pacific Ocean (PO) along a latitudinal transect (170°W, from the western and central Pacific Ocean to the Bering Sea and the Arctic Ocean) using high-throughput sequencing data, with the aim of attributing their biogeography to different ecological processes using Vellend's framework (14). As the largest and deepest ocean on Earth, the PO is an ideal

place to study how ecological processes shape phytoplankton biogeography. First, phytoplankton communities in the pelagic PO are more stable than those in coastal and estuarine waters, so the short-term temporal variations can be insignificant and ignored, which is a premise for calculating spatial turnover and ecological processes in a metacommunity constructed by spatial community assembly (16, 17). Second, the PO harbors various distinct ecological provinces whose geographical and hydrological conditions have been well studied (27). A large-scale study across the PO (e.g., along a latitude transect) allows a high coverage of species diversity and environmental gradients, which are essential for the action of selection, while ocean currents may promote the contribution of dispersal-related processes (22, 28).

We focused on three phytoplankton groups (i.e., *Synechococcus*, diatoms, and haptophytes) because they are diverse, abundant, widespread, and ecologically important phytoplankton in the ocean, with distinct traits that may influence the contribution of different ecological processes in determining their communities (29). *Synechococcus* are pico-sized ($<2$ $\mu$m) prokaryotes, while diatoms and haptophytes are larger eukaryotes, with sizes ranging from nanophytoplankton (2 to 20 $\mu$m) to microphytoplankton ($>20$ $\mu$m); together they cover the full size spectrum of phytoplankton. Diatoms are autotrophic r strategists with high growth rate, while most haptophytes are K strategists with low growth rate and the ability to shift between autotroph and heterotroph (i.e., mixotroph) (30, 31). Previous studies also showed that microbial communities could be driven by different processes with varying relative importance at different spatial scales (32, 33). We hypothesize that contribution of assembly processes in phytoplankton communities may also vary at different spatial scales and that a finer scale (e.g., a climatic zonal scale, compared to a basin scale) could provide better explanations for their latitudinal biogeography. Therefore, we asked (i) whether the relative importance of ecological processes in phytoplankton communities estimated from the climatic zonal scale is different from that determined from the basin scale, (ii) whether different phytoplankton groups are regulated by the same or different ecological processes, and (iii) what factors control the balance of niche-neutral processes in phytoplankton communities.

## RESULTS

**Biogeography of phytoplankton communities.** After quality control and sequence filtering, communities were built for diatoms, *Synechococcus*, and haptophytes, respectively (Table S2). The rarefaction curves were drawn by showing the number of detected operational taxonomic units (OTUs) with increasing number of subsampled sequences. They were all saturated or nearly saturated for samples in local, climatic zonal, and basin scales, showing a good coverage of species richness of the three phytoplankton groups in our study (Fig. S2). Haptophyte OTUs had the highest site occupation frequency (percentage of sites at which an OTU occurred), 0.28, followed by *Synechococcus* (0.084) and diatoms (0.068) (Fig. S3). All *Synechococcus* OTUs had a limited site occupation frequency below 0.4.

We described the community structure differences (i) between surface and deep chlorophyll maximum (DCM) layers and (ii) among climatic zones (including subarctic, subtropical, and tropical areas) using Bray-Curtis dissimilarity. We found significant zonal differences in the OTU structure in all three types of phytoplankton, as shown in the nonmetric multidimensional scaling (NMDS) plots (analysis of similarity [ANOSIM], $P < 0.001$) (Fig. 1b). Zonal differences were much larger than layer differences (Fig. 1c). Notably, *Synechococcus* communities were more similar to each other in the subarctic area than that in subtropical and tropical areas, while haptophyte communities showed the opposite pattern, indicating different dominant processes within phytoplankton groups at same place.

**Contribution of ecological processes to phytoplankton communities at basin and zonal scales.** We found a significant distance decay relationship (DDR) for all the phytoplankton groups from both surface and DCM layers, reflecting the spatial variations of phytoplankton communities caused by ecological processes (Fig. 2a). Community similarity of haptophytes (38.13%, on average) was much higher than those of diatoms (11.06%, on average) and *Synechococcus* (9.57%, on average). A similar pattern was found in the DCM samples. We found significant phylogenetic signal (positive correlation) at relatively shorter phylogenetic distances (e.g., $<0.2$) for all the phytoplankton groups (Fig. S4), suggesting the applicability to

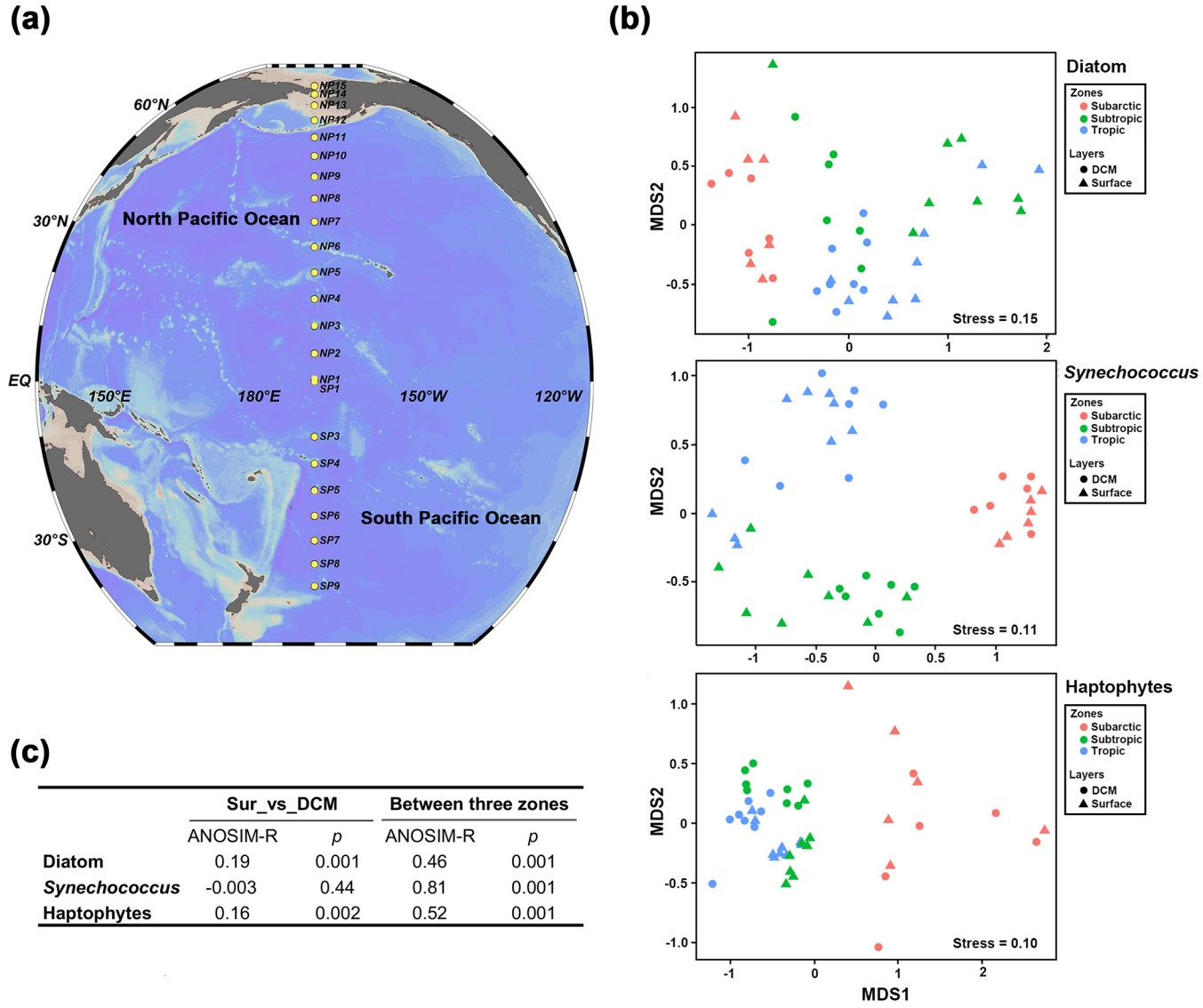

**FIG 1** Sampling stations and biogeographic distribution pattern of diatom, *Synechococcus*, and haptophyte communities. (a) Location of sites plotted by Ocean Data View (v 5.0.0) (Schlitzer, Reiner, Ocean Data View, odv.awi.de, 2021); (b) NMDS of the phytoplankton communities based on Bray-Curtis dissimilarity between them. Samples were divided by layers (surface and DCM) and climatic zones (subarctic, 50°N ~ 70°N; subtropic, 23°27′N ~ 35°N and 23°27′S ~ 35°S; tropic, 23°26′N ~ 23°26′S). (c) ANOSIM testing the significant difference between divided groups of samples (by layers and climatic zones). A higher ANOSIM *R* value indicates that two groups are more different from each other.

quantify phylogenetic turnover among the closest relatives (i.e., using $\beta$-mean-nearest taxon distance [$\beta$MNTD] and the null model to calculate ecological processes) (17).

Our null model analysis showed that, at the basin scale, heterogeneous selection was the main ecological process regulating the community structure of each phytoplankton group (Fig. 2b). For the surface samples, heterogeneous selection contributed 94.37%, 63.60%, and 68.4% to the assembly processes of diatom, *Synechococcus*, and haptophyte communities, respectively, followed by ecological drift (4.33%, 19.48%, and 25.97% for diatoms, *Synechococcus*, and haptophytes, respectively). Dispersal limitation had remarkable influences on the surface community assembly of *Synechococcus* (13.42%). Similar contributions of ecological processes were found in the DCM communities. We also showed that environmental factors were more important than spatial factors in accounting for community compositions of each type of phytoplankton at both surface and DCM layers (Table S3). This was suggested by the larger effects of pure environmental factors (Env|Geo, partitioned out spatial effects from environmental effects) than pure spatial factors (Geo|Env, partitioned out environmental effects from spatial effects) on the variations in the phytoplankton community.

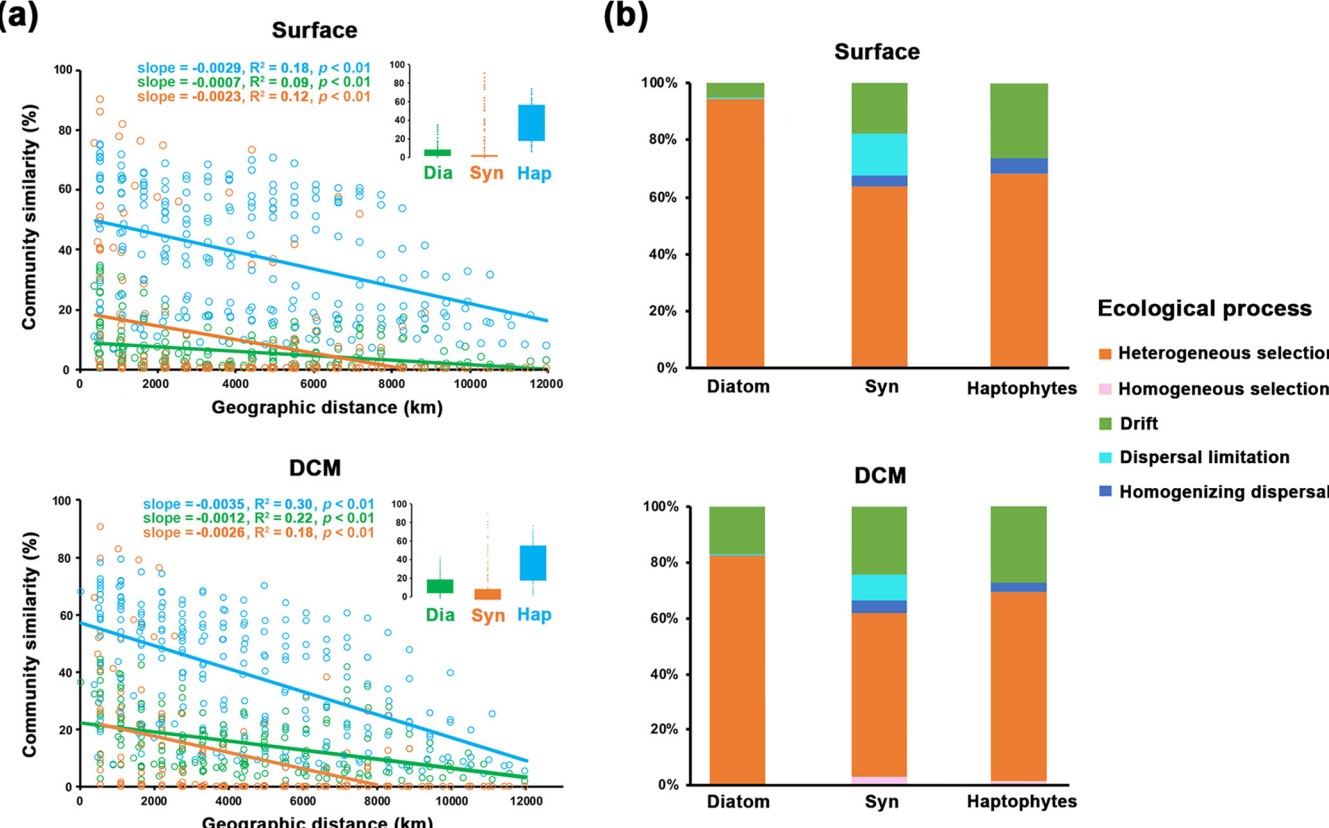

**FIG 2** Distance decay relationship (DDR) and relative contribution of ecological processes at basin scale. (a) DDRs between community similarity (Bray-Curtis) and geographic distance phytoplankton communities. Box plots of community similarity for each phytoplankton group are shown at upper right. (b) Relative importance of each ecological process in regulating the community structures of phytoplankton. For each phytoplankton group, both surface and DCM samples were analyzed. Syn, *Synechococcus*.

Notably, pure spatial factors significantly influenced *Synechococcus* communities ($P < 0.001$ for both surface and DCM), indicating an important role of dispersal limitation.

At the zonal scale, relative importance of ecological processes varied spatially and displayed different patterns among phytoplankton groups (Fig. 3a). Heterogeneous selection overwhelmingly dominated the community assembly of diatoms across all zones (93.10%, on average). From subarctic to tropic, the contribution of heterogeneous selection increased in *Synechococcus* communities (subarctic, 6.06%; subtropic, 40.00%; tropic, 59.17%) but decreased in haptophyte communities (subarctic, 95.45%; subtropic, 75.00%; tropic, 63.97%). $\beta$-Nearest taxon index ($\beta$NTI) of the 3 phytoplankton groups displayed a similar pattern (Fig. 3b). Shannon diversity ($H$) of *Synechococcus* communities increased from subarctic (0.96, on average) to tropic (2.03, on average) with reduced community similarity (averages of 62.70% and 16.75%, respectively), while haptophytes showed an opposite zonal pattern (Fig. 3c and d). For diatoms, community similarity was low across all zones (<20%) while Shannon diversity remained stable at 2.80.

The neutral community model (NCM) indicated that, from subarctic to tropic, both contribution of neutral processes ($R^2$) and immigration rate ($m$) increased in haptophyte communities but decreased in *Synechococcus* communities (see Fig. S5 in the supplemental material). Diatom communities did not fit well with the NCM ($R^2 < -0.5$). Moreover, diatom OTUs beyond the prediction threshold were mostly (>90%) below the predicted frequency, showing that they had low frequency of site occupation though their mean relative abundances were high (Fig. S6).

**Factors controlling the contribution of ecological processes in phytoplankton communities.** After controlling for other environmental variables by partial Mantel test (Table 1), we showed the following. (i) Salinity ($r = 0.30$, $P < 0.001$), euphotic layer depth (ELD)

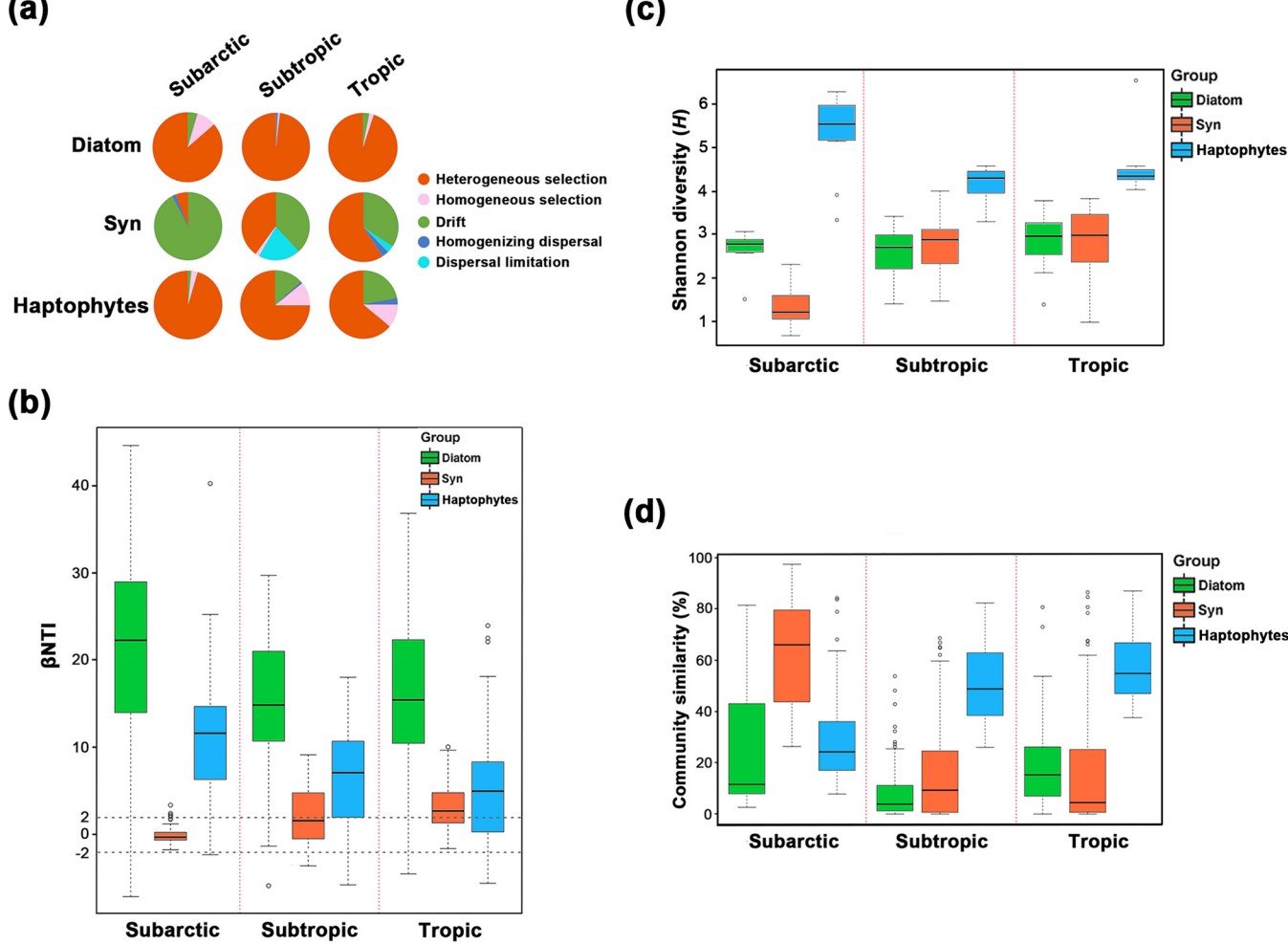

**FIG 3** Ecological processes and community structures at the zonal scale. For each phytoplankton group, samples were divided by climatic zones, with surface and DCM layers combined. (a) Relative contribution of ecological processes to the variations in phytoplankton communities calculated by the null model. (b) Values of phylogenetic turnover ($\beta$NTI) between each two samples in communities. A $\beta$NTI value below $-2$ indicates homogeneous selection; a $\beta$NTI value above 2 indicates heterogeneous selection; and a $|\beta$NTI$|$ value below 2 indicates dispersal or drift. (c) $\alpha$-Diversity (i.e., Shannon-Wiener index [$H$]) of phytoplankton communities at different zones. (d) Community similarity (Bray-Curtis) of phytoplankton at different zones. Syn, *Synechococcus*.

($r = 0.26$, $P < 0.001$) and temperature ($r = 0.25$, $P < 0.001$) contributed most to the phylogenetic turnover of diatom communities, followed by nitrate, silicate, and photosynthetically active radiation (PAR), with minor correlations ($r < 0.2$). (ii) Assembly processes of *Synechococcus* communities were mainly driven by physical factors, including temperature, ELD, salinity, PAR, and mixed-layer depth (MLD), with temperature as the most important factor ($r = 0.31$, $P < 0.001$). $\beta$NTI of *Synechococcus* was positively related to temperature difference between two samples, showing that a large difference in temperature between communities allowed heterogeneous selection to contribute more (Fig. 4a) and reduced the community similarity (Fig. 4c). Furthermore, by grouping samples by a finer temperature gradient, we showed that *Synechococcus* communities from hotter areas had larger $\beta$NTI values than those from cooler areas ($P < 0.01$, by analysis of variance [ANOVA]) (Fig. 4b) and the richness of *Synechococcus* was positively correlated with temperature (Fig. 4d). (iii) Nutrient factors (e.g., nitrate, phosphate, and silicate; $r > 0.4$, $P < 0.001$) were more important than physical factors ($r < 0.4$) in determining the assembly processes of haptophyte communities. In addition, nutrient factors were much more heterogeneous in the subarctic area (Fig. 5a) and explained more (than physical factors) in the community compositions of haptophytes but less in *Synechococcus* (Fig. 5b). Diatom communities had the largest proportion of unexplained compositional variations (86%, which cannot be explained by the two kinds of factors).

mSystems®

**TABLE 1** Correlations between environmental variables and phylogenetic turnover of phytoplankton communities[a]

| Variable | Diatom | | Synechococcus | | Haptophytes | |
|---|---|---|---|---|---|---|
| | Mantel R | Adj R | Mantel R | Adj R | Mantel R | Adj R |
| Temp | 0.33*** | **0.25*** | 0.41*** | **0.31*** | 0.51*** | **0.36*** |
| Salinity | 0.38*** | **0.30*** | 0.38*** | **0.26*** | 0.53*** | **0.37*** |
| PAR | 0.066* | **0.077* | 0.081* | **0.094* | 0.21*** | **0.25*** |
| MLD | 0.066 | 0.038 | 0.13* | **0.11* | −0.0068 | −0.078 |
| ELD | 0.28*** | **0.26*** | 0.28*** | **0.27*** | 0.52*** | **0.51*** |
| Nitrate | 0.18* | **0.14* | −0.068 | −0.1 | 0.40*** | **0.41*** |
| Phosphate | 0.12* | 0.05 | 0.067 | −0.031 | 0.57*** | **0.51*** |
| Silicate | 0.14* | **0.14* | −0.064 | −0.08 | 0.44*** | **0.48*** |
| Ammonia | 0.024 | −0.032 | −0.004 | −0.079 | 0.22* | 0.13 |
| Iron | 0.004 | −0.048 | 0.029 | −0.036 | 0.18 | 0.096 |

[a]Correlations between each environmental factor (by Euclidean distance between two sites) and community phylogenetic turnover (i.e., $\beta$NTI) were tested with the Mantel test, with R indicating the correlation index. A partial Mantel test was used to assess the relationship between phylogenetic turnover and one environmental factor after controlling for other environmental variables, which generated the adjusted correlation (Adj R). Permutation was performed 999 times, with significance shown. *, $P < 0.05$; **, $P < 0.01$; ***, $P < 0.001$. Significant adjusted R values are in bold. ELD, euphotic layer depth; MLD, mixed-layer depth; PAR, photosynthetically active radiation.

Levins' niche breadth of diatom (2.34) and *Synechococcus* (2.49) communities remained low across all zones (Fig. 6). However, the niche breadth of haptophyte communities increased markedly from subarctic (3.52) to subtropical (4.65) and tropical (5.71) areas ($P < 0.001$, by Tukey's honestly significant difference [HSD] test), such as OTUs from *Prymnesium*, *Emiliania*, and several other uncultured haptophyte species (Fig. S7).

## DISCUSSION

**Contribution of ecological processes to the marine phytoplankton communities: spatial scale matters.** In our study, heterogeneous selection dominated at the basin scale but varied greatly across climatic zones, especially in the community assembly of *Synechococcus* and haptophytes. This is in accordance with a common phenomenon, that relative contribution of ecological processes depends greatly on spatial scales, which has been widely reported but little mentioned with regard to phytoplankton (34). For instance, $\beta$-diversity of the bacterial community was found to be driven by different factors at different spatial scales, highlighting the idea that the common adopted single-scale analysis may misrepresent the true impact of each process on the biodiversity and community structures (32, 33). Methodologically, among approaches to estimating the relative contribution of ecological processes (e.g., classified by $\beta$NTI value), which is a percentage value between pairwise samples, the uneven sampling efforts at different subregions (where dominant processes may be different) will cause bias in calculating the importance of processes in the entire region.

We showed that the uneven zonal contribution of ecological processes could explain some of the biogeographic distribution patterns of phytoplankton communities. For instance, *Synechococcus* communities were more dissimilar from each other in the subtropical and tropical areas than the subarctic area, while haptophyte communities showed the opposite pattern (more dissimilar in the subarctic area). This could be explained by the varied zonal contribution of heterogeneous selection, which causes high community turnover due to the among-taxa fitness difference and leads to low community similarity (15, 20). Global model estimations showed that both present species richness and future changes in species richness, evenness, biomass, community turnover rate, and size structure have latitudinal patterns and vary greatly among climatic zones, which could indicate that the ecological processes (or their relative contributions) governing phytoplankton communities could vary across spaces (12, 13). Thus, our results suggest that a proper or multiscale analysis on the ecological processes in phytoplankton communities is necessary for a better understanding of their biogeography.

The relative importance of assembly processes can vary spatially and temporally; however, there is limited understanding of mechanisms mediating the balance of deterministic and stochastic processes, especially in phytoplankton communities (35–37). Recent studies showed that both biotic (e.g., species interaction) and abiotic (e.g., hydrodynamics) factors can

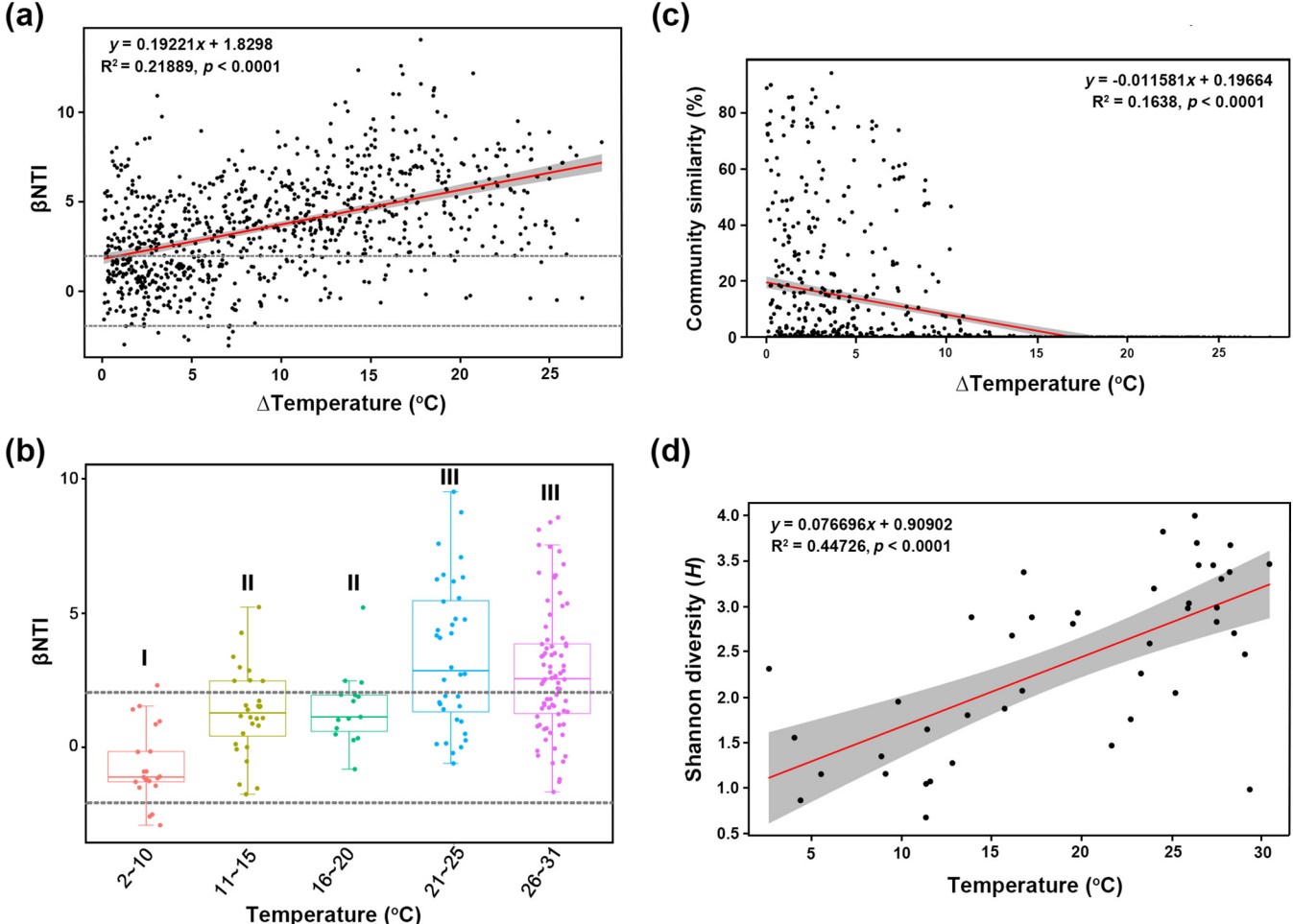

**FIG 4** Effects of temperature on phylogenetic turnover ($\beta$NTI) and structures of *Synechococcus* communities. (a) All *Synechococcus* samples ($n = 43$) were used to show the relationships between $\beta$NTI and temperature difference of each two sites. Horizontal dashed lines indicate the $\beta$NTI significance thresholds of $+2$ and $-2$. (b) $\beta$NTI and temperature. The difference in $\beta$NTI values between community assemblies was tested by ANOVA, with significantly different groups marked with different labels (i.e., I, II, and III; $P < 0.01$). (c) Community dissimilarity (Bray-Curtis) and temperature difference of two sites. (d) Shannon diversity ($H$) of a local sample and temperature. Linear regression models (red lines) and associated correlation coefficients are provided in panels a, c, and d.

affect the assembly processes of phytoplankton communities (38, 39). However, detailed relationships between specific environmental factors and relative importance of ecological processes are still unclear. Moreover, we do not know whether these relationships are different among different phytoplankton groups, which is essential to the understanding and predicting of their global distribution patterns.

**Factors driving the distinct community assembly processes of *Synechococcus* and haptophytes.** We mainly followed the trends of heterogenous selection to describe the ecological processes in phytoplankton communities in this study, although previous studies showed that the ratio of environmental selection to dispersal limitation can provide a better explanation for the microbial community than using the absolute value of contribution of each process (21, 40). In our study, dispersal limitation acted remarkably only in *Synechococcus* communities, leading to the low site occupation of species with low community similarity at the basin scale. On the other hand, ecological drift, a neutral process, contributed remarkably to the communities of all phytoplankton groups; therefore, we cannot use only the dispersal process to represent the whole neutral process. Since the contribution of selection process is negatively correlated with the contribution of neutral process in the null model (i.e., contribution of neutral = 1 − contribution of selection) and heterogeneous selection overwhelmingly dominated the selection process (i.e., the effect of homogeneous selection was negligible) in our study,

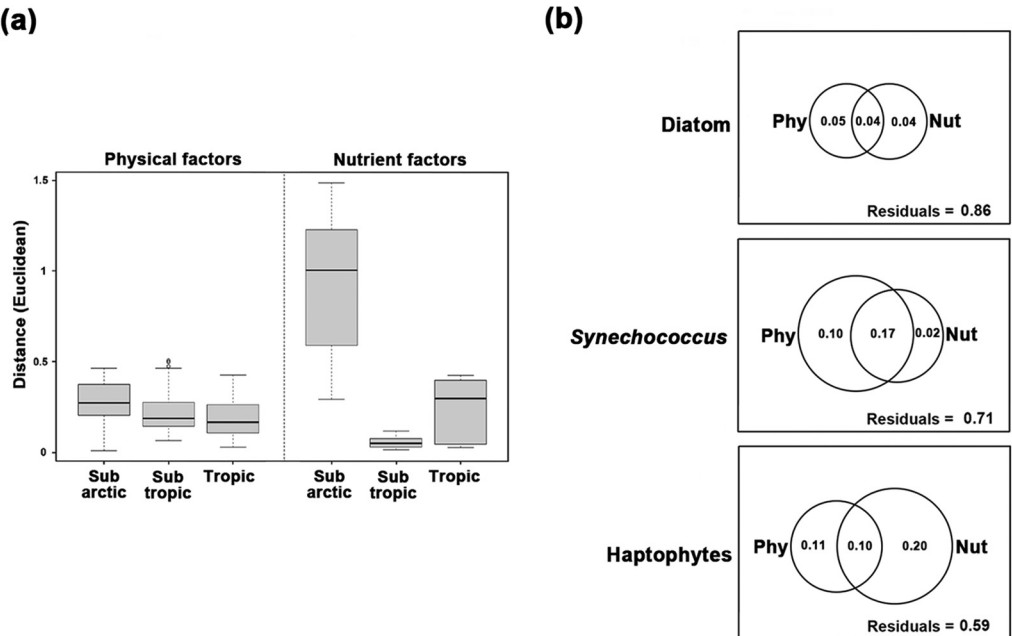

**FIG 5** Heterogeneity of physical and nutrient factors and their contributions to variations in phytoplankton communities. Environmental factors in this study were divided into two types: physical factors (Phy), including temperature, salinity, PAR, ELD, and MLD, and nutrient factors (Nut), including phosphate, nitrate, silicate, nitrite, ammonia, and iron. (a) Heterogeneity at different zones, calculated by Euclidean distance after maximum-minimum transformation. (b) Contributions to the phytoplankton community compositions, which were computed by variation partitioning analysis (VPA); the shared portion can represent a joint effect. The explanation of each kind of factor after removing the shared part is its pure contribution to the community changes. The residuals refer to the proportions of variations in community composition which cannot be explained by the two kinds of factors. Syn, *Synechococcus*.

we focused on the changes in contribution of heterogeneous selection in communities across different zones.

Assembly processes in *Synechococcus* communities have rarely been reported. In this study, we show that the relative importance of environmental selection (represented by heterogeneous selection) in *Synechococcus* varied substantially across climatic zones and was mediated by several physical factors, including temperature, salinity, ELD, and MLD. Heterogeneous selection allows more species to coexist and leads to more dissimilar structures among communities due to the environmental heterogeneity of habitats (15, 20). It contributed to the high species richness and low community similarity of *Synechococcus* communities at low latitudes in our study.

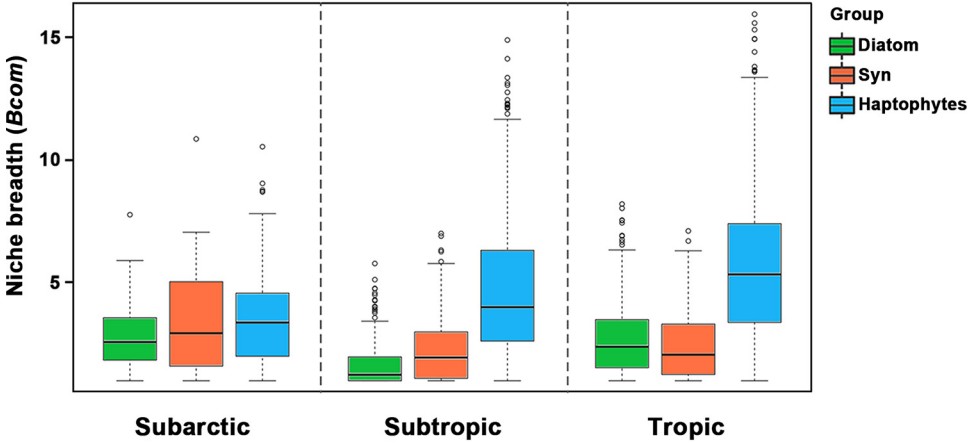

**FIG 6** Niche breadth of phytoplankton taxa across zones. Levins' niche breadth of OTUs in each phytoplankton group of communities was calculated for different climatic zones.

Temperature contributed the most to the phylogenetic turnover (i.e., $\beta$NTI) and appeared to dominate the assembly processes in *Synechococcus* communities in our study. We showed that both high temperature (i.e., high average temperature in a region) and high temperature divergence (between two sites) can cause more phylogenetic turnover in *Synechococcus* communities and lead the community assembly to be more deterministic with dominance of heterogeneous selection. Thus, our results may suggest an increasing species richness and greater community difference of *Synechococcus* in the ocean under warming effects, which would provide group-specific information for the global study of phytoplankton during climate changes (12, 13). Temperature-driven selection was also reported as the main factor shaping prokaryotic $\beta$-diversity but showed much less effect on picoeukaryotic communities in a global ocean survey (22).

We attributed the temperature-driven and zonally changing selection in *Synechococcus* communities in our study to the following causes. First, from the perspective of evolution, marine *Synechococcus* has developed strains that are phylogenetically specialized to different thermal niches and respond (e.g., growth rate) differentially due to their thermal preferences (41). The temperatures of our sampling sites ranged from 2.64°C to 30.4°C, covering the temperature niche of many *Synechococcus* clades and offering large thermal heterogeneity. Second, decline in community size may increase the importance of ecological drift, because random demographic events will matter more in a smaller population (15). The abundance of *Synechococcus* was much lower at high latitudes (e.g., subarctic) than low latitudes (e.g., subtropics and tropics) in the PO according to a previous global survey (42). This explains why ecological drift can override the effects of environmental selection in *Synechococcus* communities in the subarctic area in our study. Third, from the physiology aspect, elevated temperature may affect membrane fluidity and denature protein in *Synechococcus*, which would lead to a decrease in photosynthesis rate (41). Therefore, *Synechococcus* organisms at high temperatures face more selection effects from functional aspects, and clades (or strains) with a high level of adaptation strategies (e.g., phycobilisome-based temperature acclimation) are favored (43).

In contrast to *Synechococcus*, the relative contribution of heterogeneous selection in haptophyte communities decreased from subarctic to tropical areas, and this decrease was mainly controlled by nutrient factors. This is supported by a stronger correlation of haptophyte community structure with nutrient factors than with physical factors and suggests that the selection processes acting on haptophyte communities stemmed mainly from nutrient niche partitioning. This is understandable, as eukaryotic phytoplankton are known to be more sensitive to macronutrient availability than prokaryotes due to their larger size and faster regulation of gene expression (44). For instance, abundance of *Emiliania huxleyi* (Prymnesiophyceae, Haptophyta) showed positive correlations with macronutrient concentrations in the northeast subarctic PO, and the ratio of nitrogen to phosphate is a key factor allowing its bloom (45). Therefore, in this study, the high nutrient concentration and large nutrient heterogeneity in the subarctic area would explain the fast growth of the dominant species and the coexistence of a high number of species, which ultimately resulted in a heterogeneous selection of haptophyte communities with increased species richness and decreased community similarity. Our results would help understand how nutrient-driven bottom-up effects regulate the community dynamics of these "middle-class" nanophytoplankton (e.g., haptophytes here) whose ecological importance is often underappreciated (46).

One further note on haptophytes is that they can be mixotrophic and actively graze bacteria in oligotrophic waters (47). Previous studies suggested that mixotrophy may be common in the PO, since it is an advantageous nutritional strategy relative to autotrophy in low-nutrient oligotrophic environments, especially in low latitudes experiencing simultaneous carbon and nutrient limitation (48, 49). Eukaryotic phytoplankton, such as dinoflagellates, can utilize numerous growth strategies to survive in diverse environments (50). In our study, haptophyte OTUs had much wider niche breadth in subtropical and tropical areas than in the subarctic area. This suggested the mixotrophy of haptophytes widened the range of available resources at low latitudes, where nutrient concentrations are very low. Since organisms with wider niche breadths tend to be less influenced by environmental

filtration and can invade new habitats more easily (51), it may explain the reduced effects of environmental selection on haptophyte communities at low latitudes. In addition, the wide niche breadth of haptophytes at low latitudes, especially in tropical areas, could also be attributed to the higher dispersal rates within their communities, which led to the wide distribution of many taxa and high similarity between communities. Overall, our results showed that assembly processes in haptophyte communities were mainly controlled by nutrient factors and niche breadth of species.

**Highest selection for diatom communities but with lowest correlation with environmental factors.** Diatom communities are widely distributed and have been long known to respond quickly to environmental changes. These allow them to be used as biological indicators of environmental changes (52). In our study, diatom communities were much more phylogenetically clustered than *Synechococcus* and haptophytes, showing the overwhelming dominance of heterogeneous selection across all zones. This result is in agreement with a recent study showing the main role of environmental selection and application of phylogenetic information to interpret diatom community structure (53). Several other studies also showed that environmental selection was more important than dispersal limitation in shaping diatom communities in both ocean and freshwater (10, 53). Further, our NCM also confirmed the predominant role of environmental selection in shaping diatom communities, as the low fitness ($R^2$) implies little contribution of neutral processes (54). The lower-than-predicted frequency of site occupation indicates that their distribution was not as wide as their abundance suggested. This could be explained by the high growth rate of diatoms and the time the population needs to spread.

In agreement with previous studies, we showed that salinity and temperature each had significant positive correlations with the phylogenetic turnover of diatoms, indicating their important roles in promoting the divergence of diatom communities (55–57). In particular, while effects of salinity on diatom diversity and community compositional dynamics were often reported along a sharp salinity gradient (56, 58, 59), our results suggested that even a narrow range (30.44 to 36.15 practical salinity units [PSU]) of salinity can lead to the high diversification in diatom communities in the pelagic ocean. Unexpectedly, although we found that heterogeneous selection contributed the most to the spatial turnover of diatom communities, the correlation between environmental factors and the variations in diatom community compositions was quite low. While contribution of other unmeasured factors, such as the depth of nutricline, in our study cannot be ruled out (10), another important reason could be that most diatom species are r strategists, with high growth rate at a favorable environment. Rapid nutrient uptake by these dominant diatom species would weaken the correlation between nutrient concentrations and community structures (5, 60), which is different from K strategists (e.g., haptophytes), which have low growth rates, and r strategists (*Synechococcus*), which mainly rely on physical factors (30, 61). Therefore, direct estimation of environmental selection pressure using *in situ* environmental factors potentially leads to an underestimation with large proportion of unexplained variations, especially for r strategists relying on nutrients or other easily depleted factors (11, 53, 62).

**Synthesis.** We propose a conceptual paradigm to show the changing relative importance of ecological processes (represented by heterogeneous selection) with their controlling factors and the responses of phytoplankton communities across climatic zones in the PO (Fig. 7). In general, from the subarctic to tropical area, physical factors, especially temperature, were responsible for the increased heterogeneous selection in *Synechococcus*, leading to the higher species richness and lower community similarity at low latitudes. In addition, dispersal limitation contributed remarkably to the *Synechococcus* communities, leading to the low site occupation frequency. The selection process in haptophyte communities was controlled mainly by nutrient factors. The large spatial heterogeneity of nutrient conditions in the subarctic area led to a more heterogeneous selection, increased haptophyte species diversity, and reduced their community similarity. The mixotrophic lifestyle of haptophytes may widen their niche breadth in the oligotrophic tropical and subtropical areas, making the communities less environmentally selected. Last, diatom communities were overwhelmingly governed by heterogeneous selection in all zones, resulting in low similarity between

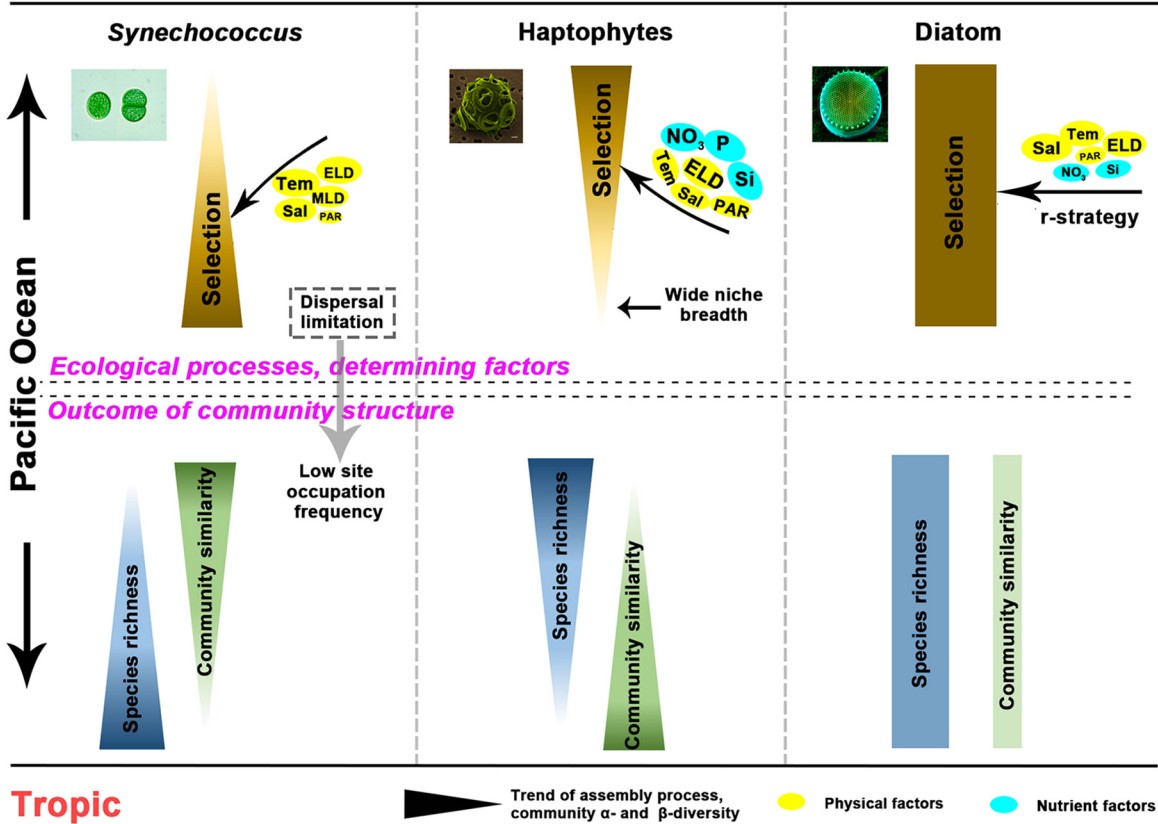

**FIG 7** Conceptual paradigm showing the zonal pattern of ecological processes regulating the phytoplankton communities in the Pacific Ocean. Ecological processes are represented by (heterogeneous) selection. The zonal trend of relative importance of ecological processes with its controlling factors (e.g., physical and nutrient factors) and variations in phytoplankton community structures (i.e., species richness and community similarity) is shown. Tem, temperature; Sal, salinity; ELD, euphotic layer depth; MLD, mixed-layer depth; PAR, photosynthetically active radiation; P, phosphate; Si, silicate; $NO_3$, nitrate.

communities. This high contribution of heterogeneous selection but weak correlation between community structure and measured *in situ* environmental parameters could be explained by the r strategy of diatom species, which is characterized by high growth rate, leading to a rapid response of community composition to environmental changes.

**Conclusions.** In this study, for the first time, we disentangled the ecological processes of three key phytoplankton groups (i.e., diatoms, *Synechococcus*, and haptophytes) simultaneously along the same gradients in the PO. We showed that heterogeneous selection, by promoting species richness and reducing similarity between communities, was the dominant process shaping the communities of each phytoplankton group at a basin scale. However, its relative importance varied greatly among different phytoplankton groups in different climate zones, explaining the uneven latitudinal distribution of $\alpha$- and $\beta$-diversity of phytoplankton in the PO. By cross-group comparison of the relative magnitude of ecological processes influencing phytoplankton community assembly, our study leads to a new hypothesis that the importance of these processes may partially differ between taxa, a finding that needs to be verified. We also highlight the importance of identifying key factors affecting the relative importance of assembly processes (or the balance between niche and neutral processes) in phytoplankton communities across spaces, which will enhance our understanding of the biogeography of phytoplankton communities in the ocean and advance our ability to predict their patterns and variability in response to climate changes.

**MATERIALS AND METHODS**

We used a road map to summarize our main analysis performed in this study (Fig. S1). In brief, phytoplankton community tables were constructed by analyzing publicly available raw sequencing data. Biogeographic

patterns were shown based on community similarity between samples and explained by ecological processes at both basin and climatic zonal scales. The relative contribution of each ecological process was estimated by the null model and supported by the results from neutral community model and variation partitioning analysis. Factors controlling the relative importance of ecological processes were analyzed based on the correlations between measured environmental factors and $\beta$NTI from the null model. Details of these analyses are described below.

**Data collection.** We used the high-throughput sequencing data from two previous studies of phytoplankton (i.e., diatoms, *Synechococcus*, and haptophytes) communities in the PO (3, 5). In brief, water samples were collected from both surface and deep chlorophyll maximum (DCM) layers along 170°W from the south PO (40°S) to the Arctic Ocean (68°N), during the KH-13-7 cruise (south PO, from December 2013 to February 2014) and KH-14-3 cruise (north PO, from June to August 2014) of the R/V *Hakuho Maru* (JAMSTEC/University of Tokyo) (Table S1; Fig. 1a). At each station, 1 L of seawater was filtered through a 0.2-$\mu$m-pore-size polycarbonate membrane with gentle vacuum, and filters were stored at −80°C. DNA extraction followed the method described in a previous study (63). Briefly, lysis buffer and glass beads were added to the vials containing the filters, and the vials were agitated with a Biospec bead beater. Then, samples were incubated at 70°C for 60 min, and the liquid phase was transferred to a 1.5-ml centrifuge tube, vortexed for 10 s, and placed on ice for 30 min. Cell debris was precipitated by centrifugation at 10,000 $\times$ *g* for 20 min, and the supernatant was transferred into a 1.5-ml tube containing 600 $\mu$l of isopropanol, incubated at room temperature for 10 min, and then centrifuged at 10,000 $\times$ *g* for 20 min. After the liquid phase was removed, the DNA pellets were washed with 600 $\mu$l of 70% ethanol, dried at 37°C, and resuspended in 100 $\mu$l of Tris-EDTA buffer. PCR amplification of the *rpoC1* gene (for *Synechococcus*) and group-specific 18S rRNA genes (for diatoms and haptophytes) were performed, and products were sequenced with an Ion Torrent PGM system.

We analyzed the raw sequences with mothur software (64). Barcodes and primers were removed from reads. Reads with an average quality score of >20 and lengths between 300 and 500 nucleotides (nt) were retained for analyses. Chimeras (artifact sequences formed by two or more biological sequences incorrectly joined together) were removed by the command chimera.uchime. After the above quality control, reads were identified by the reference sequences. Reads were denoised (removal of sequencing errors) using shhh.seqs with sigma value of 0.01. Operational taxonomic units (OTUs) were clustered at the cutoff level of 97% nucleotide identity. A shared-OTU table containing the abundance of each OTU in each sample (samples as rows and OTUs as columns) was generated using the make.shared routine. Singletons (with only 1 sequence in all samples) were removed using the command remove.rare.

Environmental factors, including sampling sites, including temperature, salinity, nitrate, phosphate, silicate, iron, ammonia, photosynthetically active radiation (PAR), mixed-layer depth (MLD), and euphotic layer depth (ELD), were collected from previous studies where detailed methods of measurement were described (5, 65). Briefly, temperature, salinity, and MLD of seawater were measured by the CTD (Sea Bird Electronics) on board. The ELD was estimated from surface chlorophyll *a* concentration using the empirical formula given by Morel et al. (66) or determined as the depth corresponding to 1% of the surface light intensity by *in situ* observation using Hyper Profiler (Satlantic). The MLD was determined by CTD profiles following the method of Suga et al. (67). Inorganic nutrient (including nitrate, phosphate, ammonia and silicate) concentrations were measured with a QuAAtro-2 continuous-flow analyzer (Bran+Luebbe, USA). The concentration of dissolved iron was measured by an automated preconcentration and introduction system (seaFAST-1 Ultra; Elemental Scientific) and high-resolution inductively coupled plasma–mass spectrometry (Finnigan ELEMENT2; Thermo Electron Corp.). Photosynthetically available radiation (PAR) data were downloaded from the observational data set of Aqua MODIS (moderate resolution imaging spectroradiometer) of the National Oceanic and Atmospheric Administration (NOAA) (https://coastwatch.pfeg.noaa.gov/erddap/griddap/erdMH1par0mday.html).

**Community similarity analysis.** We used the biogeographic distribution pattern, e.g., $\beta$-diversity, of phytoplankton communities to reflect the action of ecological processes. The following analyses were conducted using R software (v. 4.0.2) (68). First, to make a comparable analysis of diversity, communities (constructed by OTUs) of each group of phytoplankton were rarefied (i.e., subsampling) to the lowest number of the total sequence in each local community by the rrarefy function of the vegan package (69). Then, the $\beta$-diversity of communities was estimated by Bray-Curtis distance (i.e., dissimilarity) between samples using the vegdist function, and community similarity, which is equal to 1 − dissimilarity Clustering of samples based on their community similarities was shown by nonmetric multidimensional scaling (NMDS) using the monoMDS function with the first two dimensions.

To test our hypothesis that contribution of ecological processes in phytoplankton community assembly may vary at a finer scale than the basin scale, we manually divided the samples within each phytoplankton group, respectively, into three climatic zones according to their latitudes: subarctic (>40°N), subtropic (23.5 to 40°N and S), and tropic (0 to 23.5°N and S). Before estimating ecological processes, we revealed the community biogeography of each phytoplankton group by comparing community similarity (Bray-Curtis). The significant difference in community composition between two clusters (i.e., by depth layer and climatic zone) of samples was tested by analysis of similarity (ANOSIM) by the anosim function in the vegan package. A higher ANOSIM *R* value indicates a larger difference of community composition between two clusters of samples. Geographic distance between two sites was calculated by the distm function in the geosphere package, according to their latitudes and longitudes (70). Community similarity between two locales usually declines as the geographic distance between them increases, which is called distance decay relationships (DDR). It is a classic biogeographic pattern showing the spatial turnover of beta-diversity between communities and can be used to indicate the action of underlying ecological processes (71). Here, distance decay of similarity was shown as the slopes of

ordinary least-squares regressions for the relationships between geographic distance and phytoplankton community similarity of any two sites.

**Null model analysis.** To estimate the relative contribution of ecological processes to community assembly, we employed the null model based on the framework described by Stegen et al. (17). Before this model analysis, we tested for a phylogenetic signal to determine whether we could use phylogenetic turnover to make ecological inferences in our metacommunity system and to determine the most appropriate metric of phylogenetic turnover (17, 35, 72). Phylogenetic turnover here was defined as the phylogenetic distance separating OTUs found in one community from OTUs found in a second community. Using phylogenetic turnover to infer ecological processes in the assembly of communities requires a phylogenetic signal in the OTUs' optimal habitat conditions (73). The test was conducted by the mantel.correlog function (999 permutations) in the vegan package, following the procedure in a previous study (35).

For the null model analysis, phylogenetic turnover using the abundance weighed $\beta$-mean nearest taxon distance ($\beta$MNTD) metric was measured, which quantifies the mean phylogenetic distances between the two evolutionarily closest OTUs in two communities. $\beta$MNTD values higher than expected by chance indicate heterogeneous (or variable) selection in community assembly, while $\beta$MNTD values lower than expected by chance indicate homogeneous selection. The null model expectation was performed using 999 randomizations, and the deviation between the observed $\beta$MNTD and the mean of the null model distribution is shown as the $\beta$-nearest taxon index ($\beta$NTI). A significant deviation (i.e., $|\beta NTI| > 2$) indicates the dominance of selection processes: a $\beta$NTI value below $-2$ indicates significantly less phylogenetic turnover than expected (i.e., homogeneous selection) while a $\beta$NTI value above 2 indicates significantly more phylogenetic turnover than expected (i.e., heterogeneous selection) (17, 20). If the deviation is low (i.e., $|\beta NTI| < 2$), a further step is conducted to analysis whether the $\beta$-diversity of communities could be structured by dispersal or drift. In this step, Raup-Crick metric based on Bray-Curtis dissimilarity ($RC_{Bray}$) was measured and compared to the $\beta$-diversity obtained with the null model with 9,999 randomizations. $RC_{Bray}$ values above 0.95 and below $-0.95$ indicate that community turnover is driven by dispersal limitation and homogenizing dispersal, respectively, while $RC_{Bray}$ values between $-0.95$ and 0.95 indicate the dominance of drift (17, 74). Since the dominant ecological process in each paired samples can be identified by a combination of both $\beta$NTI and $RC_{Bray}$ values, the relative contribution of each ecological process in community assembly is represented by the percentage in all sample pairs. Here, we calculated the relative contribution of ecological processes in the phytoplankton communities from both basin (i.e., layer divided) and zonal (i.e., divided by climatic zones) levels.

**Mantel test and NCM.** To support the relative importance of ecological processes calculated from the null model, we supplemented the same data set with two kinds of statistical tests, each with a different focus.

First, Mantel and partial Mantel tests were used to compare the correlation between the community compositions and environmental (representing environmental selection) and spatial (representing dispersal limitation) factors for both surface and DCM layers. All environmental factors (i.e., temperature, salinity, chlorophyll $a$ level, nitrate level, phosphate level, silicate level, ammonia level, PAR, MLD, ELD, iron level, and depth) did not show normal distribution (by the shapiro.test function in R). Because some factors (e.g., temperature and nitrate level) exhibited large heterogeneity (10- to 100-fold differences) while other factors varied over small ranges (e.g., salinity), all environmental factors were $\log(x + 1)$ transformed to improve homoscedasticity and normality for multivariate statistical analysis. For spatial factors, principal coordinates of neighbor matrices (PCNM) analysis was used to generate a set of spatial variables based on the longitude and latitude coordinates of sampling stations, using the pcnm function in the vegan package (75). Before the Mantel and partial Mantel test, to avoid collinearity among factors, a further selection of both the environmental and spatial factors was conducted according to the following criteria: only factors with both a variance inflation factor (VIF) of <10 (by the vif.cca function in the vegan package) and a significant explanation of community compositions ($P < 0.05$, by the envfit function in the vegan package) were retained for downstream analysis (76). Mantel and partial mantel tests were computed by the mantel and mantel.partial functions in the vegan package.

Second, NCM was used to compare the contribution of neutral process (as opposed to selection) in phytoplankton communities across climatic zones. The NCM is based on predicting the relationship between the frequency of OTUs (i.e., site occupation frequency in the metacommunity) and their abundances across the metacommunity (54). It predicts that abundant taxa are more likely to be dispersed by chance and widespread (i.e., high frequency) in the metacommunity, while rare taxa would be lost due to drift. In the model, a high immigration rate ($m$) indicates the high species dispersal in the metacommunity (i.e., low dispersal limitation). The overall fit of the community data to the NCM is indicated as $R^2$, representing the proportion of neutral process in the metacommunity. The formulas for $m$ and $R^2$ are as follows: $Freq_i = 1 - I[1/N|N \times m \times p_i, N \times m \times (1 - p_i)]$ and $R^2 = 1 - SS_{err}/SS_{total}$, where $Freq_i$ is the occurrence frequency of OTU $i$ (a certain OTU in the metacommunity) across metacommunity; $N$ is the number of individuals per community; $m$ is the estimated immigration rate; $p_i$ is the mean relative abundance of OTU $i$ across metacommunity; $I$ is the probability density function of beta distribution; $R^2$ is the overall fit to the NCM; and $SS_{err}$ is the sum of squares of residuals and $SS_{total}$ is the total sum of squares, which come from the deviation between observed and predicted frequency. Both fitting statistics and predicted frequency of each OTU can be obtained following the R code in a previous study (77). Here, we calculated the immigration rate ($m$) and model fitness ($R^2$) for phytoplankton metacommunities from both layer and zonal scales. In particular, we plotted the observed and predicted occurrence frequency (with a 95% threshold) of diatom taxa with their mean relative abundances. Notably, $R^2$ can be a

negative value when the deviations between observed frequency and predicted frequency are large, indicating the overwhelming dominance of the deterministic process (i.e., the selection process).

**Environmental factors controlling the assembly processes and compositions of communities.** To evaluate the environmental factors that mediate the relative contribution of ecological processes in community assemblies across different climatic zones, we analyzed the correlations between all the pairwise comparisons of $\beta$NTI values and environmental factors, by Mantel and partial Mantel tests. The significance of the correlation was tested with 999 permutations, and a larger Mantel $R$ value indicates the more important role of the environmental factor in controlling the phylogenic turnover in communities (78). Due to the possible covariation of environmental factors, we also calculated the pure effects of each environmental factor on the phylogenic turnover, controlled by other environmental factors (partial Mantel test; permutations = 999). In particular, correlations between temperature (and temperature difference, by Euclidean distance) and variations in *Synechococcus* communities (including $\alpha$-diversity, $\beta$-diversity, and $\beta$NTI) were analyzed. The available environmental factors were divided into physical factors (including temperature, salinity, PAR, MLD, and ELD) and nutrient factors (including nitrate, phosphate, silicate, ammonia, and iron). Zonal differences (Euclidean distance, after maximum-minimum normalization) of both physical and nutrient factors were compared. The effects of physical and nutrient factors on phytoplankton community compositions were calculated by variation partitioning analysis (VPA), using the varpart function of vegan package. Before the VPA, selection of factors was conducted following the protocol described above for the Mantel test.

**Niche breadth estimation.** Niche breadth is another key factor influencing the relative importance of ecological processes in communities, which refers to the diversity of resources used or environments tolerated by an individual, population, species, or clade (79). An organism group with a wider niche breadth can be expected to be more metabolically flexible at the community level (51). Here, we estimated the niche breadths of taxa within phytoplankton groups from both layer and zone scales, using Levins' niche breadth ($B$):

$$B_j = 1/\sum_{i=1}^{N} P_{ij}^2$$

where $B_j$ is the habitat niche breadth of OTU $j$ in a metacommunity, $N$ is the total number of local communities in the metacommunity, and $P_{ij}$ is the proportion of OTU $j$ in local community $i$. A high $B$ value indicates that the OTU occurs widely and evenly across a large proportion of samples, representing a wide niche breadth. This calculation of niche breadth was conducted using the niche.width function in the spaa package (80). We calculated the average $B_j$ of all OTUs in a given community ($B_{com}$) as an indicator of niche breadth at the community level (21).

**Data availability.** Raw sequencing reads for analyses in this study were deposited in online open databases: for *Synechococcus*, National Center for Biotechnology Information search database (NCBI) with accession number SRP148585; for haptophytes and diatoms, DNA Data Bank of Japan (DDBJ) with accession numbers DRA004899 to DRA004901. R scripts used for the statistical analysis can be found at https://github.com/xzhimenghkust/Pacific-PP.-R-scripts.

## SUPPLEMENTAL MATERIAL

Supplemental material is available online only.
**FIG S1**, TIF file, 0.5 MB.
**FIG S2**, TIF file, 0.8 MB.
**FIG S3**, TIF file, 0.2 MB.
**FIG S4**, TIF file, 0.3 MB.
**FIG S5**, TIF file, 0.1 MB.
**FIG S6**, TIF file, 0.3 MB.
**FIG S7**, TIF file, 0.2 MB.
**TABLE S1**, DOCX file, 0.03 MB.
**TABLE S2**, DOCX file, 0.02 MB.
**TABLE S3**, DOCX file, 0.02 MB.

## ACKNOWLEDGMENTS

This study was supported by the Hong Kong Branch of Southern Marine Science and Engineering Guangdong Laboratory (Guangzhou) (SMSEGL20SC01), the Key Special Project for Introduced Talents Team of Southern Marine Science and Engineering Guangdong Laboratory (Guangzhou) (GML2019ZD0409), and the Research Grants Council of Hong Kong (16101917 and 16101318).

Z.X. and S.C. designed the study. Z.X. performed the analysis and wrote the manuscript. K.S., E.H., and X.X. provided the original data. W.W., B.C., N.H.E.H., M.L., and H.L. provided comments on the manuscript.

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
