## [Reviewer comments · mSystems]

Disentangling the ecological processes shaping the latitudinal pattern of phytoplankton communities in the Pacific Ocean

Zhimeng XU, Shunyan Cheung, Hisashi Endo, Xiaomin Xia, Wenxue Wu, Bingzhang Chen, Ngai Hei Ernest Ho, Koji Suzuki, Meng Li, and Hongbin Liu

Corresponding Author(s): Hongbin Liu, Hong Kong University of Science and Technology

Review Timeline:

Submission Date:	October 3, 2021
Editorial Decision:	November 24, 2021
Revision Received:	December 9, 2021
Accepted:	December 13, 2021

Editor: Holly Bik

Reviewer(s): Disclosure of reviewer identity is with reference to reviewer comments included in decision letter(s). The following individuals involved in review of your submission have agreed to reveal their identity: Eva Alvarez (Reviewer #2)

Transaction Report:

DOI: <https://doi.org/10.1128/mSystems.01203-21>

November 24, 2021

Dr. Zhimeng XU
Hong Kong University of Science and Technology
Ocean science
ClearWaterBay
Kowloon, None Selected
Hong Kong

Re: mSystems01203-21 (Disentangling the ecological processes shaping the latitudinal pattern of phytoplankton communities in the Pacific Ocean)

Dear Dr. Zhimeng XU:

Thank you for submitting your manuscript to mSystems. We have completed our review and I am pleased to inform you that, in principle, we expect to accept it for publication in mSystems. However, acceptance will not be final until you have adequately addressed the reviewer comments.

The two reviewers were generally positive about the scope of this study, however they have suggested a number of areas where the results and discussion could be significantly improved and clarified. Please address these concerns in a revised manuscript, providing a point-by-point response to each comment. When submitting your revised manuscript, the data availability statement should also be updated to include all accession numbers of the underlying datasets (even if these have been previously published). Currently the data statement only includes links to a GitHub repository.

Preparing Revision Guidelines

Sincerely,

Holly Bik

Editor, mSystems

Journals Department
American Society for Microbiology
1752 N St., NW

Reviewer comments:

Reviewer #2 (Comments for the Author):

Review: Disentangling the ecological processes shaping the latitudinal pattern of phytoplankton communities in the Pacific Ocean by Xu et al.

Overview

In this manuscript the authors disentangled the ecological processes that shape the communities of three key phytoplankton groups, namely *Synechococcus*, haptophytes and diatoms in the Pacific Ocean. They highlighted those environmental factors most contributing to community assembly within each group, how the determinism of the process varies with the factor itself for *Synechococcus* and haptophytes, and how diatom communities were governed by selection with minimal impact of neutral processes.

The question to be answered is clearly exposed and the interpretation of results thoughtful and supported by the data presented. I really enjoyed the Introduction and Discussion. This study is within the scope of the journal and addresses a key issue in oceanography. It has the potential to be read by many scientists of diverse fields, from theoretical ecology, experimental biology or biogeochemical modelling.

So, as I see it, the main challenge in this manuscript is to be accessible to readers that, as myself, are not familiar with the specificities of genomic data analysis. Methods and Results section are a bit hard to follow. I honestly think that investing some time in making the manuscript more approachable to non-experts in genomics would potentially make it reach a wider audience, because the main findings are really interesting. Even if that means to explain methods with a bit more detail and paying more attention to include definitions that are missing.

Major points

1. Lack of clarity in Methods section

I would suggest that the authors include a small summary or roadmap at the beginning of the methods section, to explain which are the steps of analysis performed and which is the objective of each one, or the result obtained from each if they prefer. Please, take in mind that my poor understanding of the genetic analysis, could be because my expertise in this area is too shallow, but in general, methods section has been very difficult to follow.

It is very difficult to understand the different analysis performed in the different sections. Section 2.3 is a bit clearer, in what the analysis is intended for and which is the final result of the analysis. Section 2.2., on the other hand, does not include this information and it is very difficult to understand what the authors obtained after the analysis. What is exactly a community similarity analysis and how it is obtained from genomic data?

Later in this section, authors explained how they used the Mothur software. I am not familiar with the software and maybe that's the reason why, but I do not understand many sentences in this paragraph. What does it mean to denoise a sequence? What does it mean to build a 'shared' table?

In all Methods sections, I suspect some more information would be beneficial to other readers as well. Maybe to add some information about the objective of the analysis and the information obtained afterwards would make them a bit clearer.

In Section 2.6 is not clear what is niche breadth. It seems it refers to the range of resources that a species uses, but it is not clear in the text.

2. Some concepts in Results section are not defined

There are some concepts in the results section that are not defined, and it is not clear what they refer to, some examples:

L292 It is a relatively common term, but I still think that few words about what is a rarefaction curve are necessary.

L294 What is 'site occupation frequency'?

L309 What is 'distance decay relationship'?

Minor points

Methods

L91 remove 's' from consideration.

L140 and L153 Although methods are the same as previous studies, some brief explanation would be necessary to understand the type of data collected.

L165 Just a personal opinion but to specify the package used to plot seems irrelevant here.

L168 What are 'samples of each phytoplankton group'? Any sample does not contain all groups?

Results

L300 Fig. 1a does not show dissimilarity.

Discussion

L448 This explanation regarding membrane fluidity and the type and abundance of phycobilisomes is a bit confusing. How these two concepts relate to each other?

L469 This niche breadth refers to a specific type of resource or factor (e.g. nutrients) or it is a general one? Looking at equation in L280, it seems it is a general estimate. So, mainly a curiosity, it is possible to know across which range of the different factors/resources the niche breadth extends?

L516 The authors mention here that mixotrophy seems the reason for these extended niche breadth in warmer areas, an although that is for sure one option could be others, especially if in warmer areas the importance of neutral process is higher, as the authors pointed out in L474. Is there any strong reason that supports that mixotrophy is the most probable reason, and to include it in the paradigm?

Reviewer #3 (Comments for the Author):

Review attached.

Review: *Disentangling the ecological processes shaping the latitudinal pattern of phytoplankton communities in the Pacific Ocean* by Xu et al.

Overview

In this manuscript the authors disentangled the ecological processes that shape the communities of three key phytoplankton groups, namely *Synechococcus*, haptophytes and diatoms in the Pacific Ocean. They highlighted those environmental factors most contributing to community assembly within each group, how the determinism of the process varies with the factor itself for *Synechococcus* and haptophytes, and how diatom communities were governed by selection with minimal impact of neutral processes.

The question to be answer is clearly exposed and the interpretation of results thoughtful and supported by the data presented. I really enjoyed the Introduction and Discussion. This study is within the scope of the journal and addresses a key issue in oceanography. It has the potential to be read by many scientists of diverse fields, from theoretical ecology, experimental biology or biogeochemical modelling.

So, as I see it, the main challenge in this manuscript is to be accessible to readers that, as myself, are not familiar with the specificities of genomic data analysis. Methods and Results section are a bit hard to follow. I honestly think that investing some time in making the manuscript more approachable to non-experts in genomics would potentially make it reach a wider audience, because the main findings are really interesting. Even if that means to explain methods with a bit more detail and paying more attention to include definitions that are missing.

Major points

1. Lack of clarity in Methods section

I would suggest that the authors include a small summary or roadmap at the beginning of the methods section, to explain which are the steps of analysis performed and which is the objective of each one, or the result obtained from each if they prefer. Please, take in mind that my poor understanding of the genetic analysis, could be because my expertise in this area is too shallow, but in general, methods section has been very difficult to follow.

It is very difficult to understand the different analysis performed in the different sections. Section 2.3 is a bit clearer, in what the analysis is intended for and which is the final result of the analysis. Section 2.2., on the other hand, does not include this information and it is very difficult to understand what the authors obtained after the analysis. What it is exactly a community similarity analysis and how it is obtained from genomic data?

Later in this section, authors explained how they used the Mothur software. I am not familiar with the software and maybe that's the reason why, but I do not understand many sentences in this paragraph. What does it mean to denoise a sequence? What does it mean to build a 'shared' table?

In all Methods sections, I suspect some more information would be beneficial to other readers as well. Maybe to add some information about the objective of the analysis and the information obtained afterwards would make them a bit clearer.

In Section 2.6 is not clear what is niche breadth. It seems it refers to the range of resources that a species uses, but it is not clear in the text.

2. Some concepts in Results section are not defined

There are some concepts in the results section that are not defined, and it is not clear what they refer to, some examples:

L292 It is a relatively common term, but I still think that few words about what is a rarefaction curve are necessary.

L294 What is 'site occupation frequency'?

L309 What is 'distance decay relationship'?

Minor points

Methods

L91 remove 's' from consideration.

L140 and L153 Although methods are the same as previous studies, some brief explanation would be necessary to understand the type of data collected.

L165 Just a personal opinion but to specify the package used to plot seems irrelevant here.

L168 What are 'samples of each phytoplankton group'? Any sample does not contain all groups?

Results

L300 Fig. 1a does not show dissimilarity.

Discussion

L448 This explanation regarding membrane fluidity and the type and abundance of phycobilisomes is a bit confusing. How these two concepts relate to each other?

L469 This niche breadth refers to a specific type of resource or factor (e.g. nutrients) or it is a general one? Looking at equation in L280, it seems it is a general estimate. So, mainly a curiosity, it is possible to know across which range of the different factors/resources the niche breadth extends?

L516 The authors mention here that mixotrophy seems the reason for these extended niche breadth in warmer areas, although that is for sure one option could be others, especially if in warmer areas the importance of neutral process is higher, as the authors pointed out in L474. Is there any strong reason that supports that mixotrophy is the most probable reason, and to include it in the paradigm?

Xu et al. Disentangling the ecological processes shaping the latitudinal pattern of phytoplankton communities in the Pacific Ocean.

Comments to the Authors

This study by Xu et al. seeks to understand how ecological processes determine phytoplankton community composition along a latitudinal gradient through the Pacific Ocean. Importantly, this study encompasses 3 important phytoplankton groups, *Synechococcus*, diatoms and haptophytes, that span the 3 major size classes. The authors found that different ecological factors played taxonomic-specific roles in controlling a diversity, which also differed depending on latitude.

This study has the potential to be a high-impact, well cited paper, as the results are of interest to the broad oceanography community as well as those involved in global ecosystem modelling. The study is well written, but I think the manuscript can be improved in a few different ways.

Firstly – a smaller scale focus (eg comment in response to Line 170) would provide useful data to researchers that study the Southern Pacific Ocean vs the interest and research already undertaken in the Northern Pacific. Further subdividing the climatic zones would enable a closer comparison with literature (Ocean Station Papa, Geotraces transects) and would provide a more robust discussion.

Secondly – I foresee this manuscript being well-received by oceanographers, yet the discussion lacks references to the growing body of oceanography studies in the Pacific ocean. Lab studies would also do well to be referenced here, and I'm sure there are many more relevant references for controls on phytoplankton growth than the reference to soil bacteria (Line 401-405).

Line 73: You should acknowledge that in addition to the direct role of temperature on driving the present and future patterns in phytoplankton, the indirect impacts of temperature (changes in stratification, changes in grazing behavior related to temperature, etc – for further details, read Benedetti et al. (2021) and Henson et al. (2021)) are critical influences.

Line 102: Acknowledge the extent/portions of the Pacific the transect runs through, describe the basin characteristics (onshore/open ocean etc).

Line 116: In addition to being diverse, abundant, and widely spread, you should highlight that *Synechococcus*, diatoms, and haptophytes are ecologically important.

Line 119: You write: “*Synechococcus* are autotrophic prokaryotes with cell size less than 2 μm while diatoms and haptophytes are eukaryotes with larger size”. Using the classical (0.2-2 μm , 2-20 μm , >20 μm ; Sieburth et al., 1978) size classification scheme, the three groups actually fall into the three sizes of picophytoplankton, nanophytoplankton, and microphytoplankton. You should acknowledge this as covering the full spectrum as it is an important attribute of your study.

Line 170: Have you looked at the results if you separate the subtropic region by hemisphere? Do the different circulation patterns in the Northern subtropic Pacific Ocean and the Southern subtropic Pacific Ocean influence the findings?

Line 217: Can you explain why you chose to use a $\log(x + 1)$ transformation for your multivariate statistical analysis instead of other methods such as direct standardization or normalization of the data? Some of the readers may find this information valuable if it was included in the methods.

Line 242: Define OTU_i.

Line 315: Please clarify the logic behind interpreting the phylogenetic signal at shorter phylogenetic distances and the jump to phylogenetic turnover. Species turnover is a topic of interest to the community at this moment, but this discussion could benefit from enhanced clarity, as it is currently unclear how these are linked.

Line 353: The high r of salinity in contributing to phylogenetic turnover of diatom communities warrants more discussion. I was surprised as to why salinity would be such a significant factor unless the change in salinity is reflecting changes in the density field in which case this should be acknowledged, particularly since temperature is another influential environmental variable on density and including both may be redundant.

Line 359. Please clarify the language. Is temperature inducing heterogeneous selection?

Line 368: Consider moving figure S6 into the main text. I think this is a worthwhile figure that should not be buried in the supplemental.

Lines 396-399: Phytoplankton community turnover in response to environmental parameters is of interest to the community and should be discussed further. Key references include Benedetti et al. (2021) and Henson et al. (2021).

Lines 401-405: The discussion of the impact of environmental factors on soil microbial communities seems unnecessary. There are many lab studies, or even environmental studies that show how different chemical and physical variables influence phytoplankton growth. Paul and Bach 2020, <https://doi.org/10.1111/nph.16806>; or Coello-Camba and Agustí 2017, <https://doi.org/10.3389/fmars.2017.00168>, are good examples of meta-analyses. Also something like Laws, McClellan, Passow 2020, <https://doi.org/10.1111/jpy.13048>, could be appropriate.

Line 431: The implications of your temperature findings in relation to phylogenetic turnover are important and warrant further discussion in relation to projected climate change impacts (again, see Benedetti et al. (2021) and Henson et al. (2021)).

Line 438: Again, the reference to literature on soil microbial communities seems unnecessary.

Line 446: Please clarify your language around the role of *Synechococcus* abundance at lower latitudes and the results of ecological drift. Are you suggesting an ecological mechanism associated with *Synechococcus* abundance or a limitation to the model based on *Synechococcus* biogeography?

Lines 453-466: The importance of this portion of the discussion could be strengthened by referencing recent work by Laurie Juranek and Angelique White (Juranek, White, et al., 2020) which suggests that the ecological importance of nanophytoplankton (such as your

haptophytes) is underappreciated. Your paper presents important findings that compliment the idea of the important phytoplankton “middle class” well.

Line 468: The findings in relation to mixotrophy are significant and should be brought up earlier, perhaps included in the abstract. Your findings surrounding metabolic fluidity have important implications warranting they are highlighted more. Additionally, this section would benefit from some discussion about previous mixotrophy studies in the Pacific (see Cohen et al., 2021).

Line 521: The discussion of r-strategies in diatoms should be balanced by an acknowledgement of the k-strategies in *Synechococcus*.

Table 1: Include the (%I₀) for euphotic zone used and the kg m⁻³ threshold used for mixed layer depth.

Line 755: Bold (d) for consistency with (a-c).

Other References:

Benedetti, F., Vogt, M., Elizondo, U.H., Righetti, D., Zimmerman, N.E., and N. Gruber. 2021. Major restructuring of marine plankton assemblages under global warming. *Nature Comm.* 12, doi.org/10.1038/s41467-021-25385-x.

Cohen, N.R., McIlvin, M.R., Moran, D.M. *et al.* Dinoflagellates alter their carbon and nutrient metabolic strategies across environmental gradients in the central Pacific Ocean. *Nat Microbiol* 6, 173–186 (2021). <https://doi.org/10.1038/s41564-020-00814-7>.

Juranek, L.W., White, A.E., Dugenne, M., Henderikx Freitas, F., Dutkiewicz, S., Ribalet, F., Ferron, S., Armbrust, E.V., and D.M. Karl. 2020. The importance of the phytoplankton “middle class” to ocean net community production. *Glob. Biogeochem. Cycles* 34, doi.org/10.1029/2020GB006702.

Henson, S.A., Cael, B.B., Allen, S.A., and S. Dutkiewicz. 2021. Future phytoplankton diversity in a changing climate. *Nature Comm.* 12. Doi.org/10.1038/s41467-021-25699-w.

Sieburth, J.M.C.N., Smetacek, V., and J. Lenz. 1978. Pelagic ecosystem structure: heterotrophic compartments of the plankton and their relationship to plankton size fractions. *Limnol. Oceaogr.* 23, 1256-63.

Reponses to Editor and Reviewers

Response to Editor:

The two reviewers were generally positive about the scope of this study, however they have suggested a number of areas where the results and discussion could be significantly improved and clarified. Please address these concerns in a revised manuscript, providing a point-by-point response to each comment. When submitting your revised manuscript, the data availability statement should also be updated to include all accession numbers of the underlying datasets (even if these have been previously published). Currently the data statement only includes links to a GitHub repository.

Author response:

We are grateful for the comments provided by the editor and two reviewers. We carefully addressed these concerns and gave point-by-point response to each comment here and made corresponding corrections in the “Marked-Up Manuscript” file as well.

We updated the data availability statement by adding the accession number of raw sequencing data in our revised manuscript. Please see Line 850-853 in the Marked-Up Manuscript.

Response to Reviewer #2:

Overview

In this manuscript the authors disentangled the ecological processes that shape the communities of three key phytoplankton groups, namely *Synechococcus*, haptophytes and diatoms in the Pacific Ocean. They highlighted those environmental factors most contributing to community assembly within each group, how the determinism of the process varies with the factor itself for *Synechococcus* and haptophytes, and how diatom communities were governed by selection with minimal impact of neutral processes.

The question to be answer is clearly exposed and the interpretation of results thoughtful and supported by the data presented. I really enjoyed the Introduction and Discussion. This study is within the scope of the journal and addresses a key issue in oceanography. It has the potential to be read by many scientists of diverse fields, from theoretical ecology, experimental biology or biogeochemical modelling.

So, as I see it, the main challenge in this manuscript is to be accessible to readers that, as myself, are not familiar with the specificities of genomic data analysis. Methods and Results section are a bit hard to follow. I honestly think that investing some time in making the manuscript more approachable to non-experts in genomics would potentially make it reach a wider audience, because the main findings are really interesting. Even if that means to explain methods with a bit more detail and paying more attention to include definitions that are missing.

Author response:

To make our manuscript clear and reach a wider audience, we added more details in the Methods and Results sections and a roadmap figure to show the logistics of our analyses. Please see the details of corrections in the following responses.

Major points

1. Lack of clarity in Methods section

I would suggest that the authors include a small summary or roadmap at the beginning of the methods section, to explain which are the steps of analysis performed and which is the objective of each one, or the result obtained from each if they prefer. Please, take in mind that my poor understanding of the genetic analysis, could be because my expertise in this area is too shallow, but in general, methods section has been very difficult to follow.

Author response:

To make it clear for readers and according to your suggestion, we added a roadmap at the beginning of the methods section. We briefly described our analysis pipelines, which would make it easier for readers to follow our manuscript (See Line 141-147).

The roadmap is shown below, and we put it as Figure S1 in the supplementary file.

It is very difficult to understand the different analysis performed in the different sections. Section 2.3 is a bit clearer, in what the analysis is intended for and which is the final result of the analysis. Section 2.2., on the other hand, does not include this information and it is very difficult to understand what the authors obtained after the analysis. What it is exactly a community similarity analysis and how it is obtained from genomic data? Later in this section, authors explained how they used the Mothur software. I am not familiar with the software and maybe that's the reason why, but I do not understand many sentences in this paragraph. What does it mean to denoise a sequence? What does it mean to build a 'shared' table?

Author response:

As shown above, we drew a roadmap to show the main task and logistics of each analysis.

Section 2.2 is about the biogeographic patterns of different phytoplankton groups, which is a kind of beta-diversity analysis. Because the mechanisms underlying these biogeographic patterns are some ecological processes, it leads to our aims of quantifying the relative contributions of these processes. Here, we defined a community similarity as the compositional difference at OTU level between any two communities. We used Bray-Curtis distance to calculate the dissimilarity between two communities and the similarity equals to $1 - \text{dissimilarity}$. OTUs were obtained by clustering of sequences at 97% nucleotide identity, which is similar to the species level. Then, we got the community composition data for each sample and used them for community similarity analysis. We clarified these in the revised manuscript. See Line 197-206.

To let readers easily understand how we used Mothur to process the raw data, we added more details in the method part, see Line 168-177.

Denoise a sequence here refers to a computational method for removing sequence errors from amplicon reads.

A “shared” table here means a table with sample names as rows and OTUs as columns. For each phytoplankton group, PCR was done for each sample with primer containing unique barcode to distinguish samples. Then, amplicons of all samples were pooled together and sequenced at the same time. We clustered these sequences into OTUs, together with the abundance of each unique OTU in each sample. Lastly, we got a “shared” OTU table with sample as rows and OTUs’ abundances as columns. A “shared” OTU table can be treat as a metacommunity (a set of interacting local communities) and used for downstream analysis of ecological processes.

In all Methods sections, I suspect some more information would be beneficial to other readers as well. Maybe to add some information about the objective of the analysis and the information obtained afterwards would make them a bit clearer.

Author response:

According to your suggestion, we added more detailed information in the Methods section. We included the objectives at the beginning of all analysis sections. Particularly, we added the objective in Section 2.2.

See Line 197-199.

In Section 2.6 is not clear what is niche breadth. I seems it refers to the range of resources that a species uses, but it is not clear in the text.

Author response: Niche breadth refers to the diversity of resources used or environments tolerated by an individual, population, species, or clade (Carscadden et

al., 2020). We added this definition in the text (see Line 327-329).

Reference:

Carscadden K A, Emery N C, Arnillas C A, et al. Niche breadth: causes and consequences for ecology, evolution, and conservation. *The Quarterly Review of Biology*, 2020, 95(3): 179-214.

2. Some concepts in Results section are not defined

Author response: we have added detail definitions of some concepts in Results section. For example, the definition of “rarefaction curve”, “site occupation frequency”, “distance decay relationship”, “Env|Geo”, “Geo|Env”. See corrections in the Results section.

There are some concepts in the results section that are not defined, and it is not clear what they refer to, some examples:

L292 It is a relatively common term, but I still think that few words about what is a rarefaction curve are necessary.

Author response:

A rarefaction curve is produced by repeatedly re-sampling the pool of N individuals, at random, plotting the average number of species represented by 1, 2 ... N individuals or samples.

We have added the definition of this at Line 348-349.

L294 What it is 'site occupation frequency'?

Author response:

Site occupation frequency means the percent of sites (in total sites) that an OTU occurred. For example, if there are 10 samples (or sites) at one region, OTU1 occurred in 5 samples while OTU2 occurred in all 10 samples. Then the site occupation frequency is 0.5 for OTU1 and 1 for OTU2.

We added definition of this at Line 352.

L309 What it is 'distance decay relationship'?

Author response:

The distance decay states that the community similarity between two locales declines as the geographic distance between them increases. It is a commonly found biogeographic pattern showing the spatial turnover of beta-diversity between communities. Here in our study, we used distance decay pattern to indicate the action of underlying ecological processes.

We added the definition of distance decay relationship in the method section 2.2. See Line 218-222.

Minor points

Methods

L91 remove 's' from consideration.

Author response: Removed at Line 97.

L140 and L153 Although methods are the same as previous studies, some brief explanation would be necessary to understand the type of data collected.

Author response:

The two lines are about DNA extraction and measurement of environmental factors. According to your suggestions, we added some brief descriptions of them in the Methods section. See Line 157-164 and Line 182-194.

L165 Just a personal opinion but to specify the package used to plot seems irrelevant here.

Author response:

We agree with the suggestion and have deleted the citation of using R package “ggplot2”. See Line 205.

L168 What are 'samples of each phytoplankton group'? Any sample does not contain all groups?

Author response:

Here we mean samples within each phytoplankton group, and it (dividing samples based on climatic zones) was done for each phytoplankton group, respectively, rather than containing all groups.

To make it clear, we changed the original sentence as:

“...we manually divided the samples within each phytoplankton group, respectively...”

See Line 209-210.

Results

L300 Fig. 1a does not show dissimilarity.

Author response:

Yes, Fig. 1a is just a location of sampling sites (already mentioned in the beginning of methods section). We deleted this symbol in Line 358.

Discussion

L448 This explanation regarding membrane fluidity and the type and abundance of phycobilisomes is a bit confusing. How these two concepts relate to each other?

Author response:

In this part (Line 503-530), we explained why selection effects in *Synechococcus* communities increased with temperature from physiological aspect. We mainly used the findings from two published papers: Mackey et al., 2013 and Pittera et al., 2014. In general, high temperature affects membrane fluidity and denature protein of *Synechococcus*, which would eventually lead to the inactivation of photosynthesis. Thus, *Synechococcus* under high temperature are facing higher selection effects (than under low temperature) from functional aspects, and clades (or strains) with high levels of adaption strategies (e.g., phycobilisome (PBS)-based temperature acclimation) are favored (Mackey et al. 2013).

For instance, as said from the abstract of Mackey et al. 2013: “*Synechococcus* sp. WH8102, growth at higher temperatures led to an increase in the abundance of PBS pigment proteins, as well as higher abundance of subunits of the PSII, photosystem I, and cytochrome b6f complexes. This would allow cells to increase photosynthetic electron flux to meet the metabolic requirement for NADPH during rapid growth.”

We acknowledge that it is not clear to use “the type and abundance of phycobilisomes” here, which is usually linked to light acclimation. To avoid confusion, we corrected it as: “Therefore, *Synechococcus* under high temperature are facing more selection effects from functional aspects and clades (or strains) with high levels of adaption strategies (e.g., phycobilisome-based temperature acclimation) are favored (Mackey et al. 2013).” See Line 528-530.

L469 This niche breadth refers to a specific type of resource or factor (e.g. nutrients) or it is a general one? Looking at equation in L280, it seems it is a general estimate. So, mainly a curiosity, it is possible to know across which range of the different factors/resources the niche breadth extends?

Author response:

The niche breadth (Levins’) here refers to a general one (estimation), rather than a specific type of resources or factor. According to the equation, generally, species (OTUs here) with wide niche breadth are that occur widely and evenly across a large proportion of samples. Thus, it is an inference from species abundance distribution to their general niche breadth and cannot be mapped to ranges of specific factors/resources. Moreover, we cannot rule out that some key factors are unmeasured.

L516 The authors mention here that mixotrophy seems the reason for these extended niche breadth in warmer areas, although that is for sure one option could be others, especially if in warmer areas the importance of neutral process is higher, as the authors pointed out in L474. Is there any strong reason that supports that mixotrophy is the most probable reason, and to include it in the paradigm?

Author response:

In this paragraph, we attributed the less importance of environmental selection in

haptophyte communities at tropics to their mixotrophic strategy. It was mainly suggested by the wider niche breadth of haptophytes at tropics from our data, and these taxa (e.g., species within *Prymnesium*, *Emiliana*) with wider niche breadth were well reported to have the ability to be heterotrophic and main bacterial grazers at tropical oligotrophic areas.

Because we didn't have measured or experimental data to confirm their in situ trophic modes, the explanation of mixotrophic modes to wide niche breadth and further to the less environmental selection was a speculation; we didn't have strong evidence to support that mixotrophy is the most probable reason. To be strict, we removed the "mixotroph" symbol from our last schematic figure (Fig.7 in the Marked-up Manuscript).

Response to Reviewer #3:

Comments to the Authors

This study by Xu et al. seeks to understand how ecological processes determine phytoplankton community composition along a latitudinal gradient through the Pacific Ocean. Importantly, this study encompasses 3 important phytoplankton groups, *Synechococcus*, diatoms and haptophytes, that span the 3 major size classes. The authors found that different ecological factors played taxonomic-specific roles in controlling a diversity, which also differed depending on latitude.

This study has the potential to be a high-impact, well cited paper, as the results are of interest to the broad oceanography community as well as those involved in global ecosystem modelling. The study is well written, but I think the manuscript can be improved in a few different ways.

Firstly – a smaller scale focus (e.g., comment in response to Line 170) would provide useful data to researchers that study the Southern Pacific Ocean vs the interest and research already undertaken in the Northern Pacific. Further subdividing the climatic zones would enable a closer comparison with literature (Ocean Station Papa, Geotraces transects) and would provide a more robust discussion.

Author response:

We agree with the reviewer's idea that a smaller scale focus, e.g., between S and N Pacific Ocean (PO), will provide useful data to researchers that study Pacific Ocean and may provide a more logistic and robust discussion to readers for the present study.

However, due to imbalanced sampling efforts in our data (lack of high latitude samples at Southern PO), there would be unequal and incorrect comparison of alpha and beta diversity between Southern PO and Northern PO. As we can see from the nMDS map in Figure 1b and Figure 3c&d, the diversity and communities of all three phytoplankton groups at subarctic area were very unique from other areas.

We have analyzed our data following the reviewer's suggestion. The results can be seen **from the figure below** if we separate our samples by Southern and Northern PO. We found there were no significant difference between two semi-POs, except for *Synechococcus* (ANOSIM $R = 0.15$, $p = 0.005$), but the difference was much lower than zonal difference (ANOSIM $R = 0.81$, $p = 0.001$). The main purpose of this study was to use ecological processes to explain the phytoplankton community biogeography. It's clear that the climatic zonal distribution pattern is much significant than that between two semi-POs. Thus, we didn't add the analysis of comparison between Southern and Northern PO in the present study.

Secondly – I foresee this manuscript being well-received by oceanographers, yet the discussion lacks references to the growing body of oceanography studies in the Pacific Ocean. Lab studies would also do well to be referenced here, and I’m sure there are many more relevant references for controls on phytoplankton growth than the reference to soil bacteria (Line 401-405).

Author response:

Thanks for the suggestion!

In this paragraph, we aimed to show that factors controlling the relative importance of ecological processes in phytoplankton communities are little studied. According to the suggestion, we added some relevant discussion of findings from previous studies of phytoplankton, with focuses on how factors controlling the balance of ecological processes and some research gaps.

We corrected it as below, please see Line 468-476.

“Relative importance of assembly processes can vary spatially and temporally, however, there is limited understanding of mechanisms mediating the balance of deterministic and stochastic processes, especially in phytoplankton communities (Chen et al., 2019; Dini-Andreote et al. 2015; Rojo 2021). Recent studies showed that both biotic (e.g., species interaction) and abiotic factors (e.g., hydrodynamics) can affect the assembly processes of phytoplankton communities (Klais et al. 2017; Isabwe et al. 2018). However, detailed relations between specific environmental factors and relative importance of ecological processes are still unclear. Moreover, we don’t know whether these relations are different among different phytoplankton groups, which is essential to the understanding and predicting of their global distribution patterns.”

References:

Klais, R., Norros, V., Lehtinen, S., Tamminen, T., & Olli, K. (2017). Community assembly and drivers of phytoplankton functional structure. *Functional Ecology*, 31(3), 760-767.

Isabwe, A., Yang, J. R., Wang, Y., Liu, L., Chen, H., & Yang, J. (2018). Community assembly processes underlying phytoplankton and bacterioplankton across a hydrologic change in a human-impacted river. *Science of the Total Environment*, 630, 658-667.

Dini-Andreote, F., Stegen, J. C., Van Elsas, J. D., & Salles, J. F. (2015). Disentangling mechanisms that mediate the balance between stochastic and deterministic processes in microbial succession. *Proceedings of the National Academy of Sciences*, 112(11), E1326-E1332.

Chen, W., Ren, K., Isabwe, A., Chen, H., Liu, M., & Yang, J. (2019). Stochastic processes shape microeukaryotic community assembly in a subtropical river across wet and dry seasons. *Microbiome*, 7(1), 1-16.

Rojo, C. (2021). Community assembly: perspectives from phytoplankton's studies. *Hydrobiologia*, 848(1), 31-52.

Line 73: You should acknowledge that in addition to the direct role of temperature on driving the present and future patterns in phytoplankton, the indirect impacts of temperature (changes in stratification, changes in grazing behavior related to temperature, etc – for further details, read Benedetti et al. (2021) and Henson et al. (2021)) are critical influences.

Author response:

According to the suggestion, we added the following sentences and related references in the manuscript, mentioning the indirect role of temperature on driving the global distribution pattern of phytoplankton (See Line 77-81):

“For instance, temperature mainly drives the present and future latitudinal patterns of marine phytoplankton both directly (e.g., enhancing speciation and metabolic rates) and indirectly (e.g., changing stratification, circulation and trophic interactions) ...”

Line 102: Acknowledge the extent/portions of the Pacific the transect runs through, describe the basin characteristics (onshore/open ocean etc).

Author response:

According to the suggestion, we added the portions of the Pacific Ocean where our transect runs through and the basin characteristics which are all pelagic stations (See Line 108-110):

“In this study, we characterized the phytoplankton community structures in the pelagic Pacific Ocean along a latitudinal transect (170° W, from the western and central PO to the Bering Sea, and the Arctic Ocean) using high throughput sequencing data...”

Line 116: In addition to being diverse, abundant, and widely spread, you should highlight that *Synechococcus*, diatoms, and haptophytes are ecologically important.

Author response:

We added the highlight of their ecological importance by:

“We focused on three phytoplankton groups (i.e., *Synechococcus*, diatoms and haptophytes) because they are diverse, abundant, widespread and ecologically important phytoplankton in the ocean...”. Please see Line 123.

Line 119: You write: “*Synechococcus* are autotrophic prokaryotes with cell size less than 2 μm while diatoms and haptophytes are eukaryotes with larger size”. Using the classical (0.2-2 μm , 2-20 μm , >20 μm ; Sieburth et al., 1978) size classification scheme, the three groups actually fall into the three sizes of picophytoplankton, nanophytoplankton, and microphytoplankton. You should acknowledge this as covering the full spectrum as it is an important attribute of your study.

Author response:

We agree with your idea and according to suggestion, we added the acknowledgement that the 3 groups of phytoplankton covered the full spectrum of pico-, nano- and micro-sized phytoplankton. We rephrased our sentences as:

“*Synechococcus* are pico-sized (< 2 μm) prokaryotes while diatoms and haptophytes are eukaryotes with larger size ranging from nanophytoplankton (2 – 20 μm) to microphytoplankton (> 20 μm); together they cover the full size spectrum of phytoplankton.”

See line 125-128.

Line 170: Have you looked at the results if you separate the subtropic region by hemisphere? Do the different circulation patterns in the Northern subtropic Pacific Ocean and the Southern subtropic Pacific Ocean influence the findings?

Author response:

We analyzed the community pattern of each phytoplankton group at both north and south hemisphere of the subtropic region in the Pacific Ocean (PO). By showing the community similarity pattern in the **NMDS figure below** and testing by ANOSIM (between two hemispheres), we found there was no significant difference of phytoplankton community compositions between northern PO and southern PO ($p > 0.05$). As our main aim was to use ecological processes to explain the distinct biogeography among regions, we didn't further separate the samples within subtropic regions.

We believed that the different circulation patterns in the two sides of PO will have different effects on the community diversity and processes. However, it is hard to link them as we didn't have measured data related to these current circulations. Moreover, due to the limited sample number in our data, if we separate the subtropical samples into southern and northern ones, then each part could only have about 6 samples. This will largely decrease the accuracy of measuring the relative contribution of ecological processes which is based on comparisons between each pair of samples.

Line 217: Can you explain why you chose to use a $\log(x + 1)$ transformation for your multivariate statistical analysis instead of other methods such as direct standardization or normalization of the data? Some of the readers may find this information valuable if it was included in the methods.

Author response:

In this part of analysis, before transformation of the environmental factors, we tested the normality of each factor (by shapiro.test in R) and found that all factors didn't show normal distribution. It suggests that direct standardization (e.g., z-score) is not proper here. Besides, since some factors, such as temperature or nitrate, are much heterogeneous (can be 10 to 100-fold differences) than others (e.g., salinity), we chose to use $\log(x+1)$ transformation, which can improve homoscedasticity and normality, for our multivariate statistical analysis. Because some values are zero, we used $\log(x+1)$ instead of $\log(x)$.

We corrected the sentences as (see Line 266-270):

“All environmental factors (i.e., temperature, salinity, chlorophyll a, nitrate, phosphate, silicate, ammonia, PAR, MLD, ELD, iron and depth) didn't show normal distribution (by “shapiro.test” function in R). Because some factors (e.g., temperature and nitrate) exhibited large heterogeneity (10- to 100-fold differences) while other factors varied in small ranges (e.g., salinity), all environmental factors were $\log(x + 1)$ transformed to improve homoscedasticity and normality for multivariate statistical analysis.”

Line 242: Define OTU_i .

Author response:

OTU_i refers to a certain OTU in the metacommunity.
We added this definition in Line 296.

Line 315: Please clarify the logic behind interpreting the phylogenetic signal at shorter phylogenetic distances and the jump to phylogenetic turnover. Species turnover is a topic of interest to the community at this moment, but this discussion could benefit from enhanced clarity, as it is currently unclear how these are linked.

Author response:

Using phylogenetic information to infer ecological processes requires that phylogenetic distance between taxa approximate their ecological niche difference (for example, the difference in habitat requirements). When phylogenetic distance does approximate niche difference, niches are said to have ‘phylogenetic signal’ and to be ‘phylogenetically structured’ (Losos, 2008). The “phylogenetic turnover” here is defined as the phylogenetic distance separating OTUs found in one community from OTUs found in a second community. Using phylogenetic turnover to infer ecological processes in the assembly of communities requires ‘phylogenetic signal’ in OTUs’ optimal habitat conditions. We tested for phylogenetic signal to determine whether we could use phylogenetic turnover to make ecological inferences in our metacommunity system, and to determine the most appropriate metric of phylogenetic turnover.

We found significant phylogenetic signal at short phylogenetic distances (Figure S3), consistent with previous work (e.g., Dini-Andreote et al., 2015; Stegen et al., 2012; 2013). It is therefore most appropriate to quantify phylogenetic turnover among closest relatives. For this reason, we use the between-community version of the (abundance-weighted) β -mean-nearest taxon distance (β MNTD) and the downstream null model to estimate ecological processes (Stegen et al., 2012; 2013).

See our correction at Line 228-234 and 374.

References:

- Stegen, J. C., Lin, X., Konopka, A. E., & Fredrickson, J. K. (2012). Stochastic and deterministic assembly processes in subsurface microbial communities. *The ISME journal*, 6(9), 1653-1664.
- Stegen, J. C., Lin, X., Fredrickson, J. K., Chen, X., Kennedy, D. W., Murray, C. J., ... & Konopka, A. (2013). Quantifying community assembly processes and identifying features that impose them. *The ISME journal*, 7(11), 2069-2079.
- Dini-Andreote, F., Stegen, J. C., Van Elsas, J. D., & Salles, J. F. (2015). Disentangling mechanisms that mediate the balance between stochastic and deterministic processes in microbial succession. *Proceedings of the National Academy of Sciences*, 112(11), E1326-E1332.
- Losos, J. B. (2008). Phylogenetic niche conservatism, phylogenetic signal and the relationship between phylogenetic relatedness and ecological similarity among species. *Ecology letters*, 11(10), 995-1003.

Line 353: The high r of salinity in contributing to phylogenetic turnover of diatom communities warrants more discussion. I was surprised as to why salinity would be such a significant factor unless the change in salinity is reflecting changes in the density field in which case this should be acknowledged, particularly since temperature is another influential environmental variable on density and including both may be redundant.

Author response:

According to suggestion, we added more discussion about the relations between salinity, temperature and the phylogenetic turnover of diatom communities (See Line 579-584):

“In accord to previous studies, we showed that either salinity or temperature alone had significant positive correlations with the phylogenetic turnover of diatoms, indicating their important roles in promoting the divergence of diatom communities (Anderson and Rynearson, 2020; Sjöqvist et al., 2015; Saravanan et al., 2010). In particular, while effects of salinity on diatom diversity and community compositional dynamics were often reported along a sharp salinity gradient (Sjöqvist et al., 2015; Hu et al., 2016; Snoeijs et al., 1995), our results suggested that even a shallow range (30.44 - 36.15 psu) of salinity can lead to the high diversification in diatom communities in the pelagic ocean.”

We didn't add discussion of the changes of density field as we didn't have related measured data. From our results, salinity or temperature alone (from partial Mantel test) can cause significant changes in the phylogenetic turnover of diatom communities, playing as a strong selecting (in terms of heterogeneous selection) factor.

References:

- Anderson, S. I., & Rynearson, T. A. (2020). Variability approaching the thermal limits can drive diatom community dynamics. *Limnology and Oceanography*, 65(9), 1961-1973.
- Sjöqvist, C., Godhe, A., Jonsson, P. R., Sundqvist, L., & Kremp, A. (2015). Local adaptation and oceanographic connectivity patterns explain genetic differentiation of a marine diatom across the North Sea–Baltic Sea salinity gradient. *Molecular ecology*, 24(11), 2871-2885.
- Saravanan, V., & Godhe, A. (2010). Genetic heterogeneity and physiological variation among seasonally separated clones of *Skeletonema marinoi* (Bacillariophyceae) in the Gullmar Fjord, Sweden. *European Journal of Phycology*, 45(2), 177-190.
- Hu, Y. O., Karlson, B., Charvet, S., & Andersson, A. F. (2016). Diversity of pico-to-mesoplankton along the 2000 km salinity gradient of the Baltic Sea. *Frontiers in microbiology*, 7, 679.
- Snoeijs, P. (1995). Effects of salinity on epiphytic diatom communities on *Pilayella littoralis* (Phaeophyceae) in the Baltic Sea. *Ecoscience*, 2(4), 382-394.

Line 359. Please clarify the language. Is temperature inducing heterogeneous selection?

Author response:

Here we mean high temperature difference (between two sites) induced heterogeneous selection. We rephrased the sentence as:

“...high difference of temperature between communities allowed heterogeneous selection to contribute more...”

See Line 417-418.

Line 368: Consider moving figure S6 into the main text. I think this is a worthwhile figure that should not be buried in the supplemental.

Author response:

According to suggestion, we moved Figure S6 to the main text as Figure 5.

Lines 396-399: Phytoplankton community turnover in response to environmental parameters is of interest to the community and should be discussed further. Key references include Benedetti et al. (2021) and Henson et al. (2021).

Author response:

In this part, our aim was to show that a single scale of global estimation of ecological processes in phytoplankton communities is insufficient to explain their latitudinal distributions, while a fine scale estimation, such as from climatic zonal scale, performed better. According to suggestion, we added the following discussion, mainly from works of Benedetti et al. 2021 and Henson et al. 2021, to highlight the uneven latitudinal distribution pattern of present and future phytoplankton diversity and community patterns, which could also suggest that the ecological processes governing their communities vary across spaces (e.g., climatic zones).

We added the following sentences (see Line 456-460):

“Global model estimations showed that both present species richness and future changes in species richness, evenness, biomass, community turnover rate and size structure have latitudinal patterns and varied greatly among climatic zones, which also indicate that the ecological processes (or their relative contributions) governing phytoplankton communities could vary across spaces (Henson et al., 2021; Benedetti et al., 2021).”

Lines 401-405: The discussion of the impact of environmental factors on soil microbial communities seems unnecessary. There are many lab studies, or even environmental studies that show how different chemical and physical variable influence phytoplankton growth. Paul and Bach 2020, <https://doi.org/10.1111/nph.16806>; or Coello-Camba and Agustí 2017, <https://doi.org/10.3389/fmars.2017.00168>, are good examples of meta-analyses. Also something like Laws, McClellan, Passow 2020, <https://doi.org/10.1111/jpy.13048>, could be appropriate.

Author response:

We have deleted this paragraph and re-wrote it with more related references (See our previous response at the beginning). See Line 468-476.

We didn't use the two references (Paul and Bach 2020; Laws et al 2020) you mentioned here, because the key point discussed here is about what factors controlling the relative importance of ecological processes in phytoplankton communities, which was mainly estimated by phylogenetic turnover and not directly related to growth rate.

Line 431: The implications of your temperature findings in relation to phylogenetic turnover are important and warrant further discussion in relation to projected climate change impacts (again, see Benedetti et al. (2021) and Henson et al. (2021).

Author response:

According to suggestion, we added the following discussion, to link our results to climate change impacts:

“Thus, our results may suggest an increasing number of species richness and greater community difference of *Synechococcus* in the ocean under warming effects, which would provide group-specific information for the global study of phytoplankton under climate changes (Henson et al., 2021; Benedetti et al., 2021)”

See Line 507-514.

Line 438: Again, the reference to literature on soil microbial communities seems unnecessary.

Author response:

We rephrased the sentence as:

“Temperature-driven selection was also reported as the main factor shaping prokaryotic β -diversity, but showing much less effects on picoeukaryotic communities in a global ocean survey (Logares et al., 2020)”

See Line 510-512.

Line 446: Please clarify your language around the role of *Synechococcus* abundance at lower latitudes and the results of ecological drift. Are you suggesting an ecological mechanism associated with *Synechococcus* abundance or a limitation to the model based on *Synechococcus* biogeography?

Author response:

In this part, we mean that:

The abundance of *Synechococcus* was much lower at high latitudes (subarctic area) than low latitudes in the PO according to previous global surveys. As decline in community size may increase the importance of ecological drift because random demographic events will matter more on a smaller population (while high population size supports selection), it explains why ecological drift can override the effects of environmental selection in *Synechococcus* communities at the subarctic area in our study. Thus, this was an ecological mechanism associated with *Synechococcus* abundance, rather than model limitation.

To clarify this, we rephrased our sentences in Line 521-523.

Lines 453-466: The importance of this portion of the discussion could be strengthened by referencing recent work by Laurie Juranek and Angelique White (Juranek, White, et al., 2020) which suggests that the ecological importance of nanophytoplankton (such as your haptophytes) is underappreciated. Your paper presents important findings that

compliment the idea of the important phytoplankton “middle class” well.

Author response:

We cited this reference and added related discussion to show the implication of our results on the ecological importance of nanophytoplankton (i.e., Haptophyte here): “Our results would help understand how nutrient-driven bottom-up effects regulate the community dynamics of these “middle class” nanophytoplankton (e.g., haptophytes here) whose ecological importance is often underappreciated (Juraneck et al., 2020).” See Line 545-547.

Line 468: The findings in relation to mixotrophy are significant and should be brought up earlier, perhaps included in the abstract. Your findings surrounding metabolic fluidity have important implications warranting they are highlighted more. Additionally, this section would benefit from some discussion about previous mixotrophy studies in the Pacific (see Cohen et al., 2021).

Author response:

According to suggestion, we added “mixotrophic lifestyle” in the Abstract section in our manuscript. See Line 44.

We added more discussion related to mixotrophy, with reference of Cohen et al., 2021 “Previous studies suggested that mixotrophy may be common in the PO since it is an advantageous nutritional strategy relative to autotrophy in low-nutrient oligotrophic environments, especially in low latitudes experiencing simultaneous carbon and nutrient limitation (Edwards, 2019; Ward, 2019). Eukaryotic phytoplankton, such as dinoflagellates, can utilize numerous growth strategies to survive in diverse environment (Cohen et al., 2021)” See Line 550-554.

References:

Edwards, K. F. Mixotrophy in nanoflagellates across environmental gradients in the ocean. *Proc. Natl Acad. Sci. USA* 116, 6211–6220 (2019)

Ward, B. A. Mixotroph ecology: more than the sum of its parts. *Proc. Natl Acad. Sci. USA* 116, 5846–5848 (2019).

Cohen, N. R., McIlvin, M. R., Moran, D. M., Held, N. A., Saunders, J. K., Hawco, N. J., ... & Saito, M. A. (2021). Dinoflagellates alter their carbon and nutrient metabolic strategies across environmental gradients in the central Pacific Ocean. *Nature Microbiology*, 6(2), 173-186.

Line 521: The discussion of r-strategies in diatoms should be balanced by an acknowledgement of the k-strategies in *Synechococcus*.

Author response:

Synechococcus are fast-growing “r-strategy” phytoplankton with lower stability, while slow-growing eukaryotic algae (e.g., species within haptophytes) are “K-strategy”

organisms with higher stability (Margalef, 1978; Chen and Liu, 2011).

In this paragraph, we argued that, due to the r-strategy of diatoms which allow diatoms to use nutrient quickly and grow fast, there was a mismatch (i.e., underestimated correlation) between measured in situ concentration of environmental factors and diatom community structure, although we estimated strong environmental selection effects in their communities. This is different from K-strategists (e.g., haptophytes) with low growth rate or r-strategists (*Synechococcus*) mainly relying on physical factors. Therefore, direct estimation of environmental selection pressure using in-situ environmental factors potentially leads to an underestimation with large proportion of unexplained variations especially for r-strategists relying on nutrients or other easily depleted factors.

We added the above discussion in Line 591-593.

References:

- Margalef, R. Life-forms of phytoplankton as survival alternatives in an unstable environment. *Oceanol. Acta* 1, 493–509 (1978).
- Chen, B., & Liu, H. (2011). Temporal stability of marine phytoplankton in a subtropical coastal environment. *Aquatic ecology*, 45(3), 427-438.

Table 1: Include the (%I₀) for euphotic zone used and the kg m⁻³ threshold used for mixed layer depth.

Author response:

The depth of euphotic layer (ELD) was estimated from surface chlorophyll a concentration using the empirical formula given by Morel et al. or determined as the depth corresponding to 1% of the surface light intensity by in situ observation using Hyper Profiler (Satlantic). The MLD was determined by CTD profiles following the method of Suga et al.

We added this information in the method part, see line L182-186.

References

- Suga, T., Motoki, K., Aoki, Y. & Macdonald, A. M. The North Pacific climatology of winter mixed layer and mode waters. *J. Phys. Oceanogr.* 34, 3–22 (2004).
- Morel, A., Huot, Y., Gentili, B., Werdell, P. J., Hooker, S. B., & Franz, B. A. (2007). Examining the consistency of products derived from various ocean color sensors in open ocean (Case 1) waters in the perspective of a multi-sensor approach. *Remote Sensing of Environment*, 111(1), 69-88.

Line 755: Bold (d) for consistency with (a-c)

Author response: We bolded all the lower-case letters in the brackets in the figure legend.

December 13, 2021

Dr. Hongbin Liu
Hong Kong University of Science and Technology
Division of Life Science
Clear Water Bay, kowloon
Hong Kong
Hong Kong

Re: mSystems01203-21R1 (Disentangling the ecological processes shaping the latitudinal pattern of phytoplankton communities in the Pacific Ocean)

Dear Dr. Hongbin Liu:

I am satisfied that the authors have addressed all remaining reviewer concerns, and I am now happy to recommend final acceptance for this manuscript.

Your manuscript has been accepted, and I am forwarding it to the ASM Journals Department for publication. For your reference, ASM Journals' address is given below. Before it can be scheduled for publication, your manuscript will be checked by the mSystems senior production editor, Ellie Ghatineh, to make sure that all elements meet the technical requirements for publication. She will contact you if anything needs to be revised before copyediting and production can begin. Otherwise, you will be notified when your proofs are ready to be viewed.

Publication Fees:

We recognize that the video files can become quite large, and so to avoid quality loss ASM suggests sending the video file via <https://www.wetransfer.com/>. When you have a final version of the video and the still ready to share, please send it to Ellie Ghatineh at eghatineh@asmusa.org.

Sincerely,

Holly Bik
Editor, mSystems

Journals Department
Table S1: Accept
Fig. S3: Accept
Fig. S4: Accept
Fig. S6: Accept
Fig. S5: Accept
Fig. S1: Accept
Fig. S7: Accept
Fig. S2: Accept
Table S2: Accept
Table S3: Accept